# Experimental evidence that chronic outgroup conflict reduces reproductive success in a cooperatively breeding fish

Ines Braga Goncalves*, Andrew N Radford

School of Biological Sciences/Life Sciences, University of Bristol, Bristol, United Kingdom

**Abstract** Conflicts with conspecific outsiders are common in group-living species, from ants to primates, and are argued to be an important selective force in social evolution. However, whilst an extensive empirical literature exists on the behaviour exhibited during and immediately after interactions with rivals, only very few observational studies have considered the cumulative fitness consequences of outgroup conflict. Using a cooperatively breeding fish, the daffodil cichlid (*Neolamprologus pulcher*), we conducted the first experimental test of the effects of chronic outgroup conflict on reproductive investment and output. 'Intruded' groups received long-term simulated territorial intrusions by neighbours that generated consistent group-defence behaviour; matched 'Control' groups (each the same size and with the same neighbours as an Intruded group) received no intrusions in the same period. Intruded groups had longer inter-clutch intervals and produced eggs with increasingly less protein than Control groups. Despite the lower egg investment, Intruded groups provided more parental care and achieved similar hatching success to Control groups. Ultimately, however, Intruded groups had fewer and smaller surviving offspring than Control groups at 1-month post-hatching. We therefore provide experimental evidence that outgroup conflict can decrease fitness via cumulative effects on reproductive success, confirming the selective potential of this empirically neglected aspect of sociality.

*For correspondence:
ines.goncalves@bristol.ac.uk

**Competing interest:** The authors declare that no competing interests exist.

## Editor's evaluation

This paper experimentally investigates the fitness consequences of intergroup conflict in social fish. It finds that groups that face frequent territorial intrusion suffer costs in terms of both fertility and number of surviving offspring, despite behavioral compensation through increased parental care. These results provide clear and compelling evidence that intergroup conflict leads to lower fitness, and are therefore of substantial interest for understanding social evolution, where the importance of between-group competition has long been debated.

## Introduction

In social species, conspecific outsiders often challenge groups and their members for resources and reproductive opportunities (*Birch et al., 2019*; *Braga Goncalves and Radford, 2019*; *Kitchen and Beehner, 2007*; *Radford, 2008*; *Thompson et al., 2017*). This 'outgroup conflict'—conflict with one or more outsiders, of which 'intergroup conflict' (that between two groups) is a subset—is theorised to be a driving force in the evolution of territoriality, social structure, group dynamics, and cooperation (*Bowles, 2009*; *Gaston, 1978*; *Rusch, 2014*; *Wrangham, 1980*). Empirical research in non-human animals has traditionally focussed on aggressive outgroup contests, such as factors determining which individuals participate, their level of contribution, and who wins (*Arseneau-Robar et al., 2016*; *Green*

*et al., 2020*; *Kitchen and Beehner, 2007*), as well as the immediate fitness costs arising from loss of life, breeding position, or territory (*Goodall, 1986*; *Morris-Drake et al., 2022*; *Spong et al., 2008*). Recently, studies have begun to explore the effects of outgroup conflict beyond periods of active confrontation, documenting short-term behavioural changes (e.g. increased within-group affiliation) in the aftermath of single interactions with rivals (*Birch et al., 2019*; *Braga Goncalves and Radford, 2019*; *Mirville et al., 2020*; *Preston et al., 2020*; *Radford, 2008*). However, the cumulative build-up of threat posed by outsiders is also likely a potent stressor (*Eckardt et al., 2016*; *Samuni et al., 2019*) and may disrupt within-group relationships (*Anderson et al., 2020*; *Hellmann and Hamilton, 2019*), potentially generating long-term fitness consequences even in the absence of immediate direct effects from individual aggressive contests (*Braga Goncalves et al., 2022*; *Morris-Drake et al., 2022*). Three observational studies have found associations between outgroup conflict and reproductive success, beyond the direct death of offspring during aggressive interactions: greater intergroup conflict during pregnancy was associated with improved foetal survival in both crested macaques (*Macaca nigra*) and banded mongooses (*Mungos mungo*) (*Kerhoas et al., 2014*; *Thompson et al., 2017*), but with longer inter-birth intervals and reduced infant survival in chimpanzees (*Pan troglodytes verus*) (*Lemoine et al., 2020*). However, to test a causal link between outgroup conflict and reproductive success—that is to rule out potential confounding explanations associated with natural observations—experiments are needed.

Here, we use the daffodil cichlid (*Neolamprologus pulcher*), a model species for the study of sociality, to test experimentally the reproductive consequences of chronically elevated outgroup conflict. *N. pulcher* is a highly territorial, cooperatively breeding fish species native to Lake Tanganyika. In the wild, groups comprising a breeding pair and 0–20 subordinates of both sexes (*Wong and Balshine, 2011*) defend territories from predators, heterospecific competitors, and conspecific intruders (*Desjardins et al., 2008*; *Taborsky and Limberger, 1981*), with multiple small, contiguous territories often clustered together (*Taborsky, 1984*). Although individuals develop dear–enemy relationships with neighbours (*Frostman and Sherman, 2004*; *Sogawa et al., 2016*), aggressive disputes at shared borders are common (*Balshine et al., 2001*) and intruding neighbours can be attacked by all group members (*Balshine-Earn et al., 1998*). Single intrusion events are known to cause short-term changes in within-group behavioural interactions (*Braga Goncalves and Radford, 2019*; *Bruintjes et al., 2016*; *Taborsky, 1985*). Crucially, *N. pulcher* is a highly tractable experimental system—they are easily maintained in captive conditions, where groups display natural behaviour and breed regularly (*Heg and Hamilton, 2008*; *Jindal et al., 2017*; *Wong and Balshine, 2011*)—allowing controlled manipulations and detailed monitoring over extended periods.

To investigate the cumulative effect of elevated outgroup conflict on reproductive rate, investment (in eggs and parental care) and output, we simulated intrusions by neighbours at territorial borders in two long-term, laboratory experiments. We predicted that because repeated territorial intrusions are likely stressful and may destabilise social groups (*Arseneau-Robar et al., 2016*; *Morris-Drake et al., 2022*), there would be effects on breeding and investment decisions as parents are selected to respond to prevailing ecological (environmental and social) conditions (*Carlisle, 1982*; *McGinley et al., 1987*; *Roff, 1992*). For instance, organisms faced with prolonged challenging conditions may maximise survival at the cost of other functions such as reproduction (*Beldade et al., 2017*; *Jørgensen et al., 2006*; *Love and Williams, 2008*; *Schreck et al., 2001*), so we predicted a negative impact of outgroup conflict on reproductive rate in terms of spawning likelihood, latency to spawn, number of clutches produced, and inter-clutch interval. Female fish can modify egg investment depending on current intrinsic (e.g. maternal state and stress) and extrinsic (e.g. ecological) factors (*Faria et al., 2018*; *McCormick, 1998*; *Schreck, 2010*), with trade-offs between egg number and quality apparent in stressful conditions (*Faria et al., 2018*). So, we predicted an effect of outgroup conflict on egg number, size (weight and volume), and nutritional (lipid and protein) content. Parental care extends the opportunity for adjustment of reproductive investment, allowing compensation for lower offspring quality (*Carlisle, 1982*; *Clutton-Brock, 1991*) and poorer environmental conditions (*Östlund and Ahnesjö, 1998*). So, we predicted an effect of outgroup conflict on clutch visits and egg caring effort. Since maternal and early-life stress, initial egg size and quality, and parental care can all influence offspring characteristics (*Bell et al., 2016*; *Boogert et al., 2013*; *Jonsson and Jonsson, 2014*; *McCormick, 1998*), we predicted that outgroup conflict would directly or indirectly have a negative effect on reproductive output in terms of offspring survival, behaviour, and size.

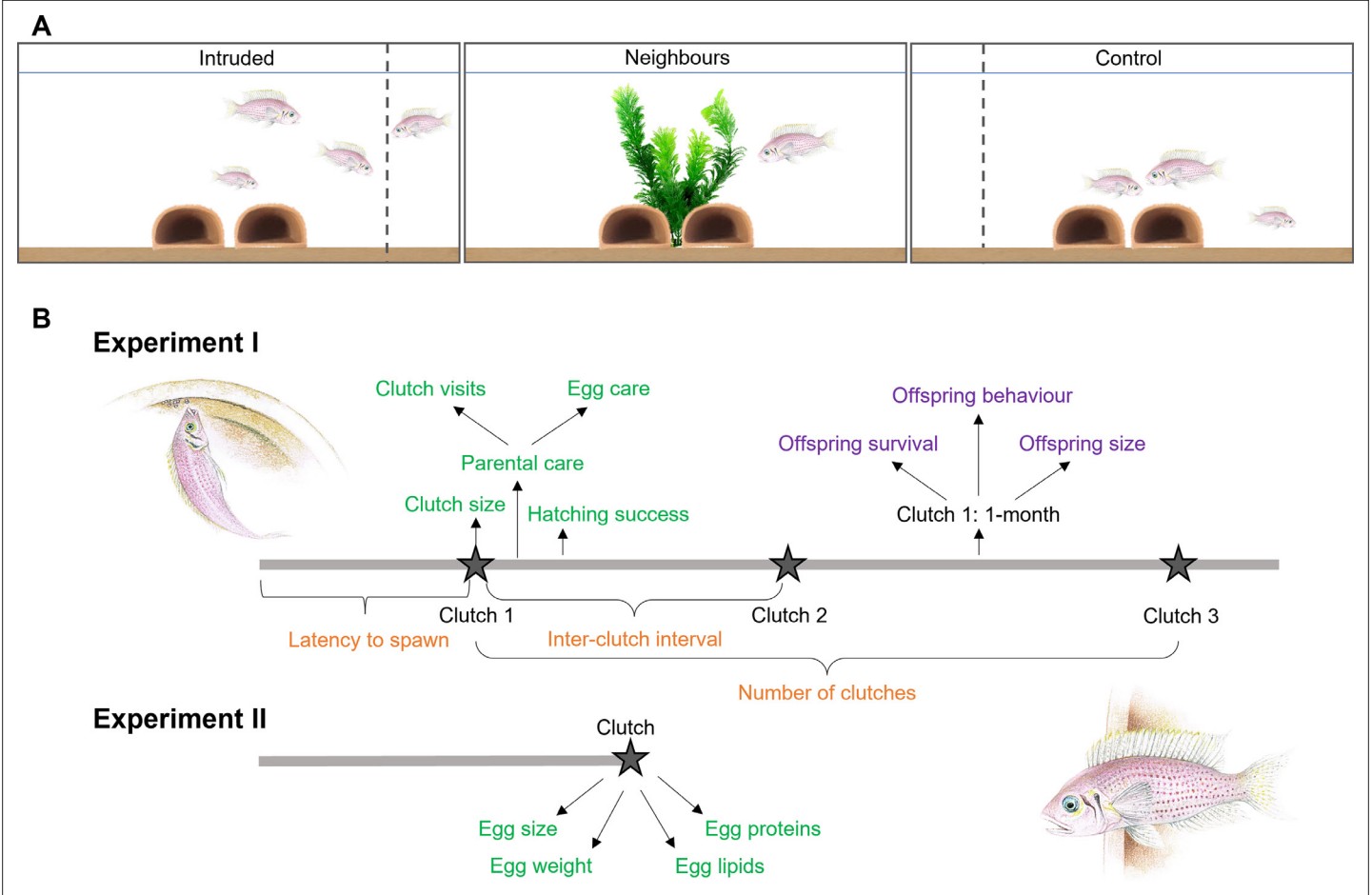

**Figure 1.** Experimental setup and timelines. (**A**) Experimental setup showing a linearly arranged tank triplet, with size-matched experimental groups (Intruded and Control) at the ends and the tank with the neighbour pair at the centre. The female neighbour is presented in the Intruded group tank to illustrate an intrusion along the border of the resident group's territory. (**B**) Timelines show measures of general reproductive behaviour (orange), egg and parental-care investment (green), and reproductive output (purple).

## Results

In both our experiments (which used different fish), 30 tanks with breeding shelters were arranged linearly in triplets: the end 'focal' tanks (one for each of two treatments: 10 'Intruded' and 10 'Control' tanks per experiment) contained a dominant pair and a subordinate; the middle tank contained a dominant pair who comprised the shared neighbours used as intruders (*Figure 1A*). Within each triplet, the same-sex dominants in all tanks and the subordinates in focal end tanks were size matched (using standard body length) at the start (see Materials and methods). In Experiment I (the longer of the two experiments; see below), focal individuals in the two treatments remained size matched at the end of the study, in both standard length (paired *t*-test, dominant male [DM]: $t_9 = 0.10$, $p = 0.920$; dominant female [DF]: $t_9 = 1.67$, $p = 0.130$; Wilcoxon signed-rank test, subordinate: $V = 15.5$, $n = 12$, $p = 0.343$) and body mass (paired *t*-test, DM: $t_9 = 0.64$, $p = 0.540$; DF: $t_9 = 0.11$, $p = 0.914$; Wilcoxon signed-rank test, subordinate: $V = 18$, $n = 12$, $p = 0.156$).

Each week in both experiments, Intruded groups experienced several 10-min territorial intrusions by one of their neighbouring individuals (mean ± standard error [SE] intrusions per week, Experiment I: 4.8 ± 0.1; Experiment II: 3.9 ± 0.3); both neighbouring individuals were used for intrusions pseudo-randomly (see Materials and methods). During intrusions, the focal group and the intruder were physically separated by a transparent partition that precluded adversaries from injuring each other and cannibalism of focal group young by intruders. Focal groups typically respond to the presence of intruders with aggressive behaviours, including attacks (rams and bites) and displays (aggressive postures, frontal displays, and fast approaches); intruders reciprocate with aggressive displays

and attacks, but also with submissive postures and displays and by turning away from the focal group (**Reddon et al., 2015**; **Sopinka et al., 2009**). Control groups did not receive intrusions but did experience the same procedural disturbances as Intruded groups, in terms of the placement and removal of barriers (see Materials and methods). The aim of Experiment I was to assess the effects of chronic outgroup conflict on reproductive rate, parental care of eggs, hatching success, and offspring survival, behaviour, and size. So, following the first three intrusions (and equivalent time in Control groups), we allowed each group to raise all clutches produced over a 13-week period (**Figure 1B**). Experiment II was a complement to Experiment I, aiming to assess how chronic outgroup conflict impacts reproductive investment in terms of egg size and nutritional content (**Figure 1B**); these are measures that can only be obtained by removing clutches. We therefore collected the first clutch produced by each group following 13 days of treatment, which represents approximately half of a reproductive cycle (**Jindal et al., 2017**); the experiment ended for a focal group once that clutch was collected. For all response measures analysed, we provide in the main text the effect of treatment (Intruded and Control) and, where significant or near-significant, the interaction between treatment and treatment duration, as the factors of core interest. Full model outputs including all tested variables (significant or not) are detailed in the Supplementary files.

## Defensive behaviour

To assess whether the presence of a neighbour in their territory was perceived as an intrusion, we recorded defensive behaviour by focal groups in Experiment 1. Intruded groups exhibited 4.5 times more defensive actions than Control groups at the start of the study (Wilcoxon signed-ranked test: $V = 0$, $n = 20$, p = 0.002; **Figure 2A**); this difference persisted until the end of the experiment (4.7 times more defence; paired $t$-test: $t_9 = 4.33$, p = 0.002; **Figure 2A**). Our protocol was therefore likely to have elevated the perceived level of outgroup conflict throughout the experimental period, as intended. Within the Intruded treatment, dominant individuals but not subordinates, increased their defensive efforts between the start and the end of the study (Wilcoxon signed-ranked test, DF: $V = 4.5$, $n = 20$, adjusted-p = 0.043; paired $t$-test, DM: $t_9 = 3.50$, adjusted-p = 0.020; subordinates: $t_5 = -0.05$, p = 0.963; **Figure 2B**); this pattern was not driven by any significant changes in intruder responsiveness to the focal group ($t_9 = 0.55$, p = 0.594). Greater defensive efforts by dominant individuals at the end of the study may be a by-product of age- or size-mediated behavioural changes—a positive effect of body size on defensive efforts has been documented in at least dominant males in this species (**Ligocki et al., 2019**)—or may have been due to the presence of offspring that need safeguarding from intruders (**Dyble et al., 2019**).

## Reproductive rate

In Experiment I, repeated territorial intrusions did not significantly affect the likelihood of spawning (McNemar test: $\chi^2_1 = 0.36$, p = 0.547), the latency to first spawn (linear mixed model [LMM]: $\chi^2_1 = 1.02$, p = 0.312; **Supplementary file 1a**) or the number of clutches produced (Wilcoxon signed-rank test: $V = 15$, $N = 10$, p = 0.396). However, Intruded groups had inter-clutch intervals that were 40% longer than Control groups (LMM: $\chi^2_1 = 3.89$, p = 0.049; **Supplementary file 1b**; **Figure 3A**). This finding aligns qualitatively with previous work on *N. pulcher*, where higher neighbour densities were associated with greater spawning latencies (**Taborsky et al., 2007**), and with studies on different species documenting how various chronic stressors may negatively impact fish reproductive rates (**Ali and Wootton, 1999**; **Mileva et al., 2011**). Longer inter-clutch intervals likely result in fewer breeding attempts per season in the wild, which is why inter-birth interval is a life-history trait commonly used to assess female reproductive success across taxa (**Clutton-Brock et al., 1984**).

## Investment in eggs

In Experiment I, there was a near-significant effect of the interaction between treatment and treatment duration on clutch size (LMM, parameter estimate [PE] = 0.85, 95% confidence interval [CI] = −0.04 to 1.70, $\chi^2_1 = 3.55$, p = 0.059; **Supplementary file 2**). Against our expectation, Intruded females showed a tendency to produce larger clutches over time (i.e. with longer exposure to outgroup conflict), relative to Control females (**Figure 3B**). As conditions that reduce the probability of future reproductive success may select for increased current reproductive investment (**Carlisle, 1982**; **McGinley et al.,**

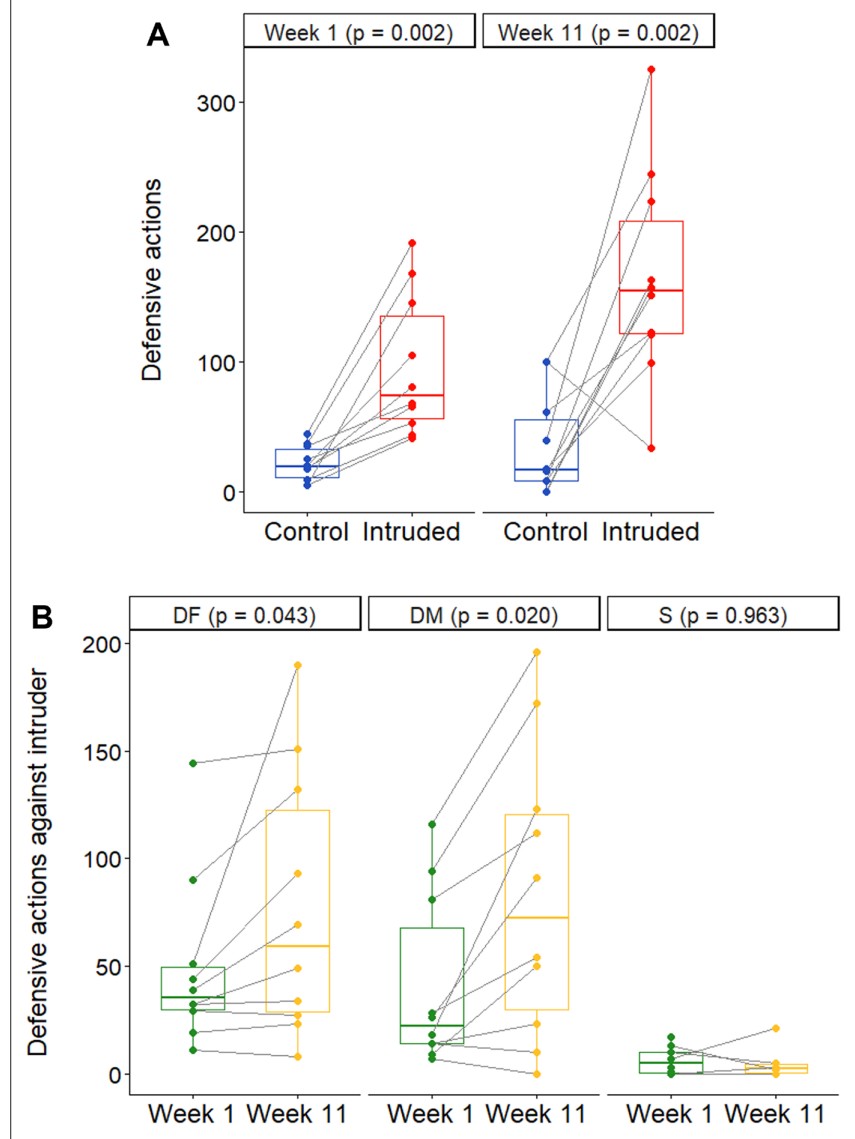

**Figure 2.** Defensive actions. (**A**) Group defensive actions displayed per 10-min trial, towards a transparent partition (blue) or the intruding female neighbour (red) in weeks 1 and 11 of Experiment I ($N = 20$ groups). (**B**) The number of defensive actions displayed by the dominant female (DF, $N = 10$), dominant male (DM, $N = 10$), and subordinate (S, $N = 6$) towards the intruding female neighbour during weeks 1 (green) and 11 (yellow) of Experiment I. Boxplots show medians, 25% and 75% quartiles, and whiskers representing 95% confidence intervals; dots are raw data, with lines connecting matched groups (**A**) or repeated measures on individuals (**B**).

The online version of this article includes the following source data for figure 2:

**Source data 1.** Number of group defensive actions displayed per 10-minute trial, towards a transparent partition (Control treatment) or the intruded female (Intruded treatment) in weeks 1 and 11 of Experiment I (n = 20 groups); and number of defensive actions displayed by the dominant females (DF, n = 10), dominant male (DM, n = 10), and subordinate (S, n = 6) per 10-minute trial, towards the intruding female neighbour during weeks 1 and 11 of Experiment I.

---

*1987*; *Olofsson et al., 2009*), it is possible that the repeated territorial invasions adversely impacted the perceived expected future reproductive success of at least the dominant females.

Experiment II allowed us to consider how increased outgroup conflict affected egg size and nutrient allocation because we removed clutches for detailed assessment. There was no significant effect of treatment on egg weight (LMM: $\chi^2_1 = 0.85$, $p = 0.356$; *Supplementary file 3a*). However, egg volume was affected by the interaction between treatment and treatment duration ($\chi^2_1 = 4.34$, $p = 0.037$;

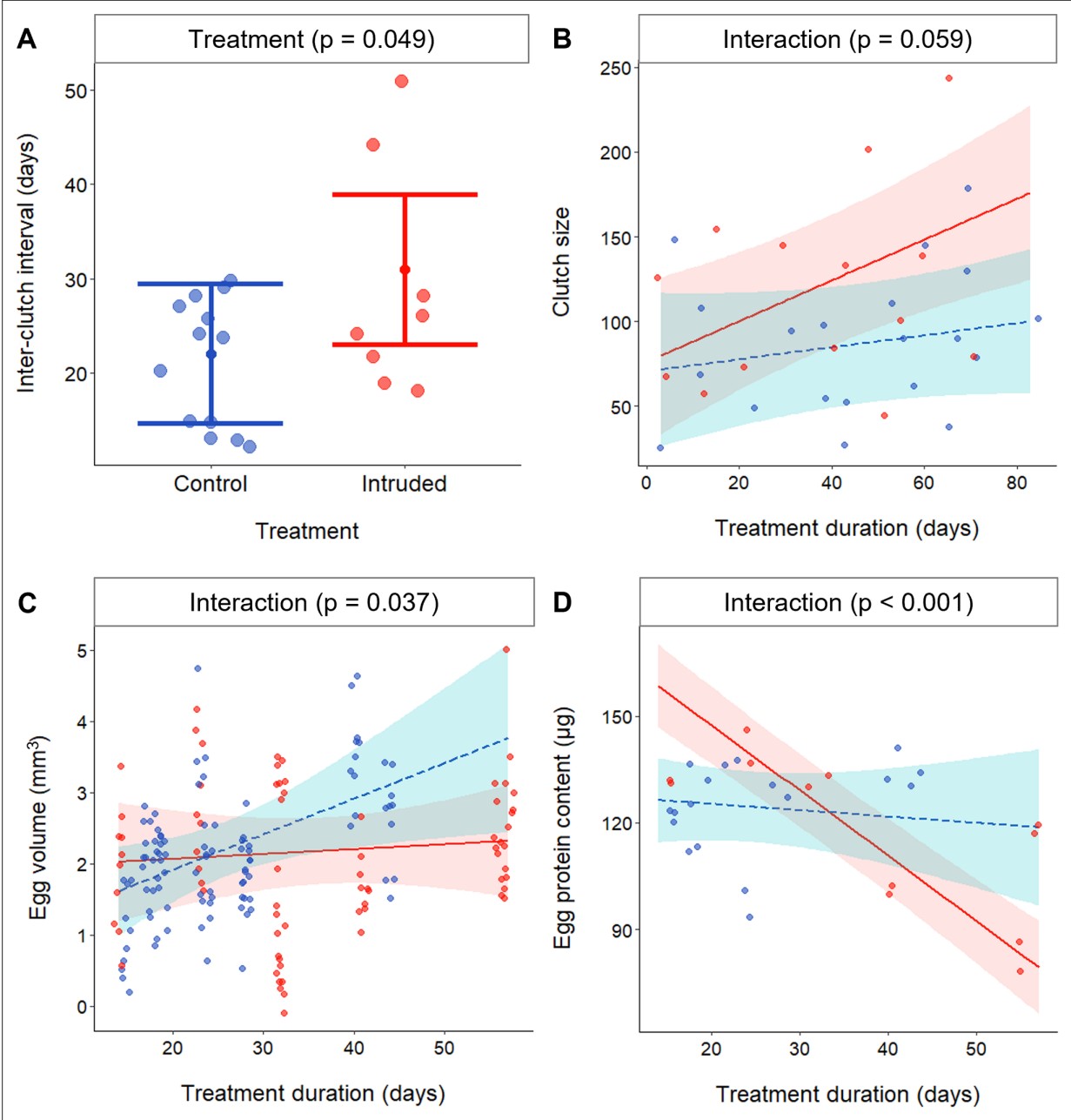

**Figure 3.** Effects of chronically elevated outgroup conflict on reproductive rate and egg investment. Effects of chronically elevated outgroup conflict (red) relative to control conditions (blue) on: (**A**) inter-clutch interval (*N* = 21 intervals); (**B**) clutch size (*N* = 34 clutches); (**C**) egg volume (*N* = 15 clutches); and (**D**) egg protein content (*N* = 15 clutches). All panels depict predicted means and associated 95% confidence intervals; dots are raw data.

The online version of this article includes the following source data for figure 3:

**Source data 1.** Inter-clutch intervals (days, n = 21 intervals); clutch size (number of eggs, n = 34 clutches); volume of eggs (mm^3, n = 15 clutches); and egg protein content (micrograms, n = 15 clutches), of clutches produced by Control and Intruded groups.

*Supplementary file 3b*; *Figure 3C*): females who took longer to spawn produced eggs with larger volume in the Control treatment (PE = 0.05, 95% CI = 0.02–0.09, $\chi^2_1$ = 7.30, p = 0.007), but there was no significant effect in the Intruded treatment (PE = 0.009, 95% CI = −0.019 to 0.037, $\chi^2_1$ = 0.51, p = 0.474). There was no significant treatment difference in egg lipid content ($\chi^2_1$ = 0.07, p = 0.797; *Supplementary file 3c*). However, egg protein content was affected by the interaction between treatment and treatment duration ($\chi^2_1$ = 18.50, p < 0.001; *Supplementary file 3d*; *Figure 3D*): mean egg protein content decreased with treatment duration in the Intruded treatment (PE = −0.97, 95% CI =

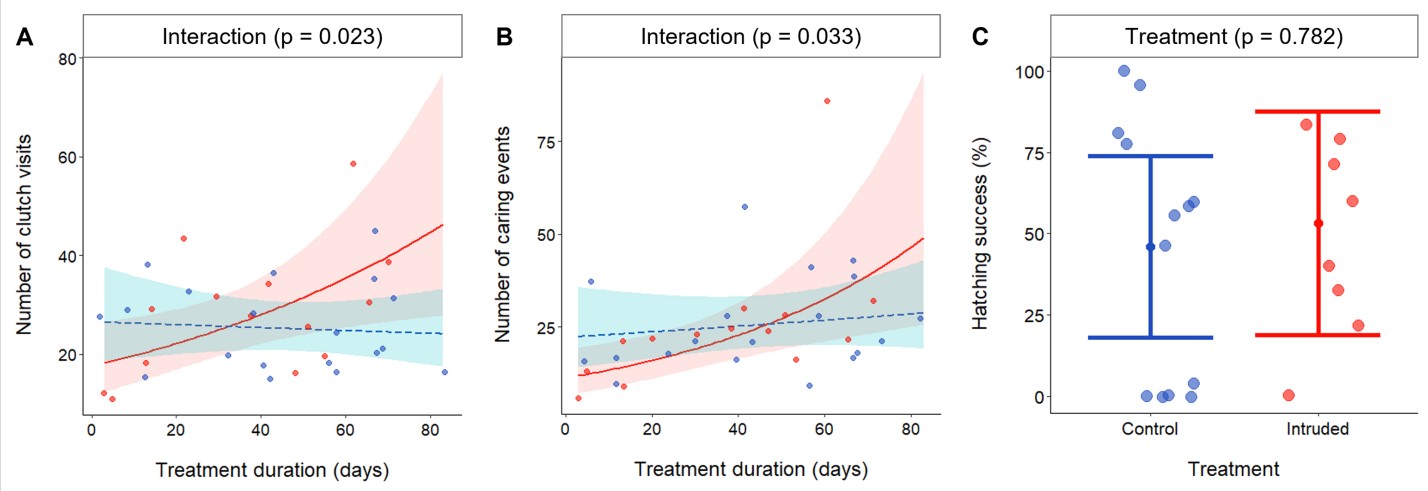

**Figure 4.** Effect of chronically elevated outgroup conflict on parental-care investment and hatching success. Effects of chronically elevated outgroup conflict (red) relative to control conditions (blue) on: (**A**) number of clutch visits (*N* = 33 clutches); (**B**) number of clutch caring (cleaning and fanning) events per 10 min observation (*N* = 33 clutches); and (**C**) offspring hatching success (*N* = 21 clutches). All panels depict predicted means and associated 95% confidence intervals; dots are raw data.

The online version of this article includes the following source data for figure 4:

**Source data 1.** Number of clutch visits (n = 33 clutches) and number of clutch caring evens (n = 33 cutches) provided to; and offspring hatching success (n = 21 clutches) of, clutches produced by Control and Intruded groups.

−1.65 to −0.30, $\chi^2_1$ = 6.30, p = 0.012), but did not change significantly in the Control treatment (PE = 0.41, 95% CI = −0.47 to 1.30, $\chi^2_1$ = 0.99, p = 0.320). Previous work on *N. pulcher* described a positive correlation between time taken to produce a clutch and mean egg size (*Taborsky et al., 2007*). Outgroup conflict thus had an increasingly (across the period of intrusions) disruptive effect on this relationship, as well as causing a progressive decline in egg quality, at least with respect to protein content. It is possible that the much higher egg concentrations of protein cf. lipids made it easier to detect an effect in just the former, but reductions in just egg protein content have also been reported in female eastern fence lizards, *Sceloporus undulatus*, exposed experimentally to elevated glucocorticoids during egg production (*Ensminger et al., 2018*). Protein is often the most abundant dry constituent of eggs (*Blaxter, 1969*), provides amino acids for tissue growth and energy via catabolic processes (*Blaxter, 1969*), and has been found to be positively correlated with fertilisation, hatching success, and early-life survival (*Kamler, 2005*).

## Investment in parental care

In Experiment I, repeated territorial intrusions had an increasingly positive effect on parental-care behaviour, as measured from the number of clutch visits (as a proxy for protection effort) and the combined number of egg-fanning and egg-cleaning events (as an estimate of caring effort) (*Supplementary file 4*). The time spent on such activities was similarly affected (*Supplementary file 5*) but we present just the results relating to the number of events here. The number of clutch visits was affected by the interaction between treatment and treatment duration (generalised linear mixed model [GLMM]: $\chi^2_1$ = 5.17, p = 0.023; *Supplementary file 4a*; *Figure 4A*): clutch visits increased over time in the Intruded treatment (PE = 0.01, 95% CI = 0.003–0.20, $\chi^2_1$ = 5.77, p = 0.016), but did not change significantly in the Control treatment (PE = −0.001, 95% CI = −0.008 to 0.005, $\chi^2_1$ = 0.13, p = 0.719). Similarly, the number of care behaviours performed was affected by the interaction between treatment and treatment duration (GLMM: $\chi^2_1$ = 4.54, p = 0.033; *Supplementary file 4b*; *Figure 4B*): caring behaviour increased over time in the Intruded treatment (PE = 0.02, 95% CI = 0.008–0.031, $\chi^2_1$ = 8.82, p = 0.003), but did not change significantly in the Control treatment (PE = 0.004, 95% CI = −0.004 to 0.011, $\chi^2_1$ = 0.85, p = 0.356). Our results contrast those of studies that have shown reductions in parental (*Vitousek et al., 2014*; *Vitousek et al., 2018*) and alloparental (*Mares et al., 2012*) care in response to immediate or short-term stressful situations but, as effects became evident over the course of the experiment, it is possible that acute and chronic stressors elicit opposing

behavioural responses. The increased parental care in the Intruded treatment may have at least partly compensated for lower relative egg investment (see above) because there was no significant treatment difference in hatching success ($\chi^2_1 = 0.08$, p = 0.782; *Supplementary file 6*; *Figure 4C*). Similar hatching success does not, however, necessarily equate to similar offspring quality (*Botterill-James et al., 2019*).

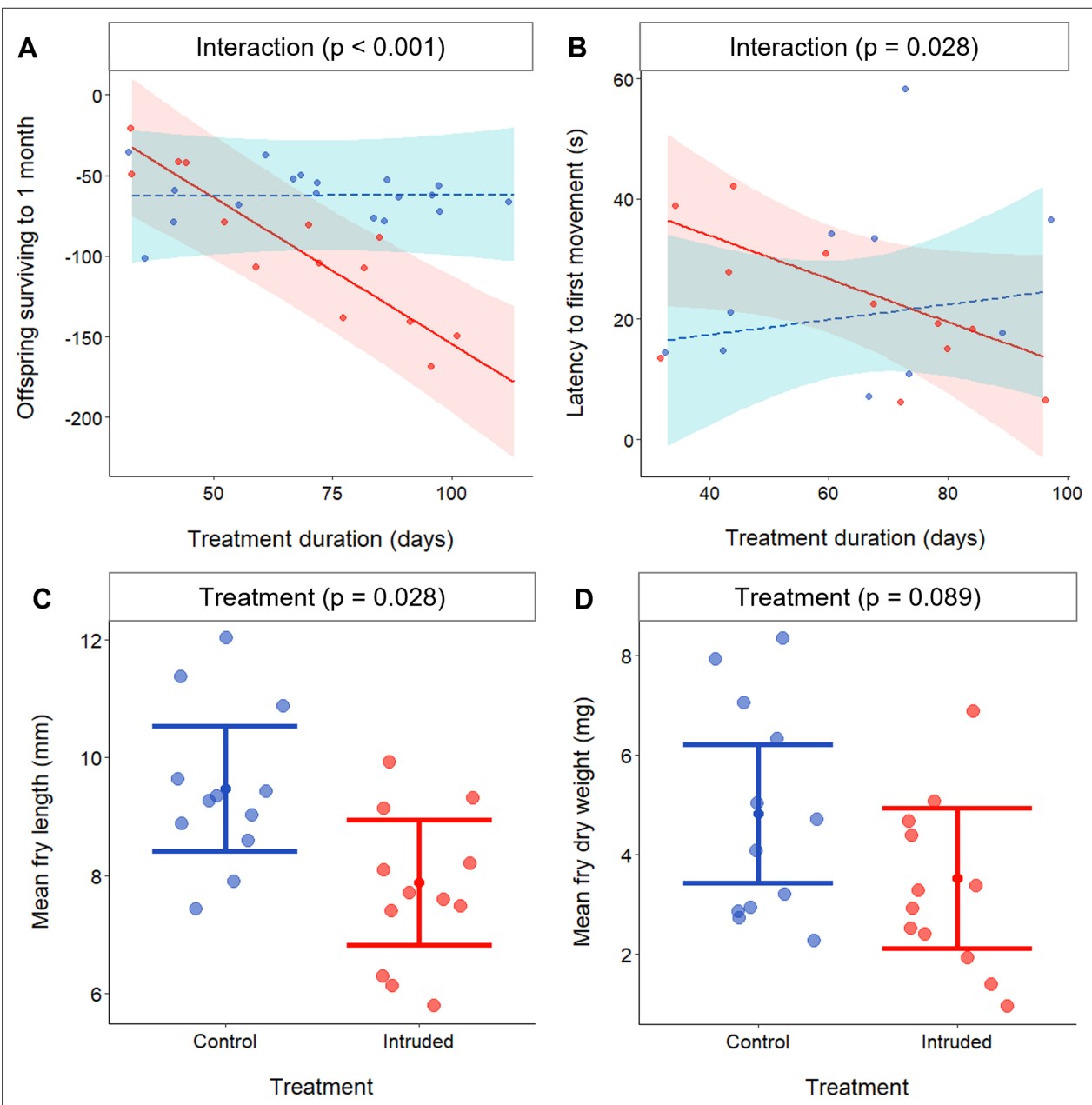

**Figure 5.** Effect of chronically elevated outgroup conflict on reproductive output. Effect of chronically elevated outgroup conflict (red) compared to control conditions (blue) on: (**A**) number of offspring surviving to 1 month (*N* = 32 clutches); (**B**) latency to first movement of offspring post-stimulus (*N* = 21 clutches); (**C**) mean offspring standard length (*N* = 24 clutches); and (**D**) mean offspring dry weight (*N* = 24 clutches). All panels depict predicted means and associated 95% confidence intervals; dots are partial residuals (**A**) or raw data (**B–D**).

The online version of this article includes the following source data for figure 5:

**Source data 1.** Number of offspring surviving to 1-month post hatching (n = 32 clutches); latency for first movement of offspring post-stimulus (n = 21 clutches) in a startling stimulus test; mean offspring standard length (n = 24 clutches); and mean offspring dry weight (n = 24 clutches), from clutches produced by Control and Intruded groups.

## Reproductive output

The absolute number of offspring surviving to 1 month in Experiment I was affected by the interaction between treatment and treatment duration (LMM: $\chi^2_1$ = 13.31, p < 0.001; *Supplementary file 7*; *Figure 5A*): the number of surviving young decreased with more exposure to outgroup conflict (Intruded groups; PE = −1.82, 95% CI = −2.50 to −0.99 $\chi^2_1$=9.70, p = 0.002), but there was no significant change over time in the Control treatment (PE = 0.01, 95% CI = −0.49 to 0.52, $\chi^2_1$ = 0.003, p = 0.956). The result was about 1.8 fewer offspring surviving to 1 month per experimental day in the Intruded treatment relative to the Control treatment. Neither food competition nor heterospecific predation can explain the lower offspring number in the Intruded treatment because all tanks were provided daily ad-lib food and there were no predators in our laboratory setting. Adults in the Intruded treatment may have displayed filial cannibalism (*Jindal et al., 2017*), but we are not aware of reports of within-group fry or juvenile cannibalism in this species. Alternatively, low early-life survival may have resulted from stress effects on parents that reduced the quality of offspring (*Anderson et al., 2020*; *Beldade et al., 2017*; *Love and Williams, 2008*; *McCormick, 1998*), stress transmission between group members that were particularly detrimental to young (*Noguera et al., 2017*), the direct early-life experience of outgroup conflict on offspring (*Jonsson and Jonsson, 2014*) or any combination of these factors.

To investigate offspring behaviour, we placed 5–10 1-month-old young from Experiment I clutches in a novel environment (20 × 20 cm container). Following a 30-min settling period, there was no significant treatment difference in offspring baseline activity level (proportion of active individuals, LMM: $\chi^2_1$ = 1.69, p = 0.194; *Supplementary file 8a*) or mean nearest-neighbour distance ($\chi^2_1$ = 0.11, p = 0.754; *Supplementary file 8b*). After this undisturbed period, we assessed the response to a startle stimulus; normally, on detecting danger, young *N. pulcher* swiftly sink to the bottom and remain motionless (*Taborsky, 1984*; *Watve and Taborsky, 2019*). Treatment had no significant effect on offspring latency to freeze immediately post-stimulus ($\chi^2_1$ = 0.06, p = 0.806; *Supplementary file 8c*). However, latency for the first offspring to move post-stimulus was affected by the interaction between treatment and treatment duration ($\chi^2_1$ = 4.84, p = 0.028; *Supplementary file 8d*; *Figure 5B*): post-stimulus latency to move decreased over time in the Intruded treatment (PE = −0.35, 95% CI = −0.68 to −0.07, $\chi^2_1$ = 5.43, p = 0.020) but there was no significant change in the Control treatment (PE = 0.29, 95% CI = −0.20 to 0.78, $\chi^2_1$ = 1.49, p = 0.223). The speed of return to activity is thought to determine vulnerability to predation, because prey movement enhances detection and attack rates by predators (*Lima and Dill, 1990*; *Middlemis Maher et al., 2013*).

Immediately following the behavioural tests, we sacrificed the offspring to assess standard length and dry weight. Compared to Control individuals, Intruded offspring were 19% shorter (LMM: $\chi^2_1$ = 4.85, p = 0.028; *Supplementary file 9a*; *Figure 5C*) and 30% lighter, though this was not statistically significant as a difference ($\chi^2_1$ = 2.89, p = 0.089; *Supplementary file 9b*; *Figure 5D*). The smaller size at 1 month of age indicates that outgroup conflict hinders early-life growth, despite unlimited access to food. Size in early life is considered a key determinant of survival (*Ahnesjo, 1992*; *Candolin et al., 2022*) due to, for instance, lower susceptibility to starvation (*Marsh, 1998*) and greater ability to escape predators (*Ahnesjo, 1992*), and of reproductive success in adulthood (*Royle et al., 2005*), although compensatory growth occurs in fishes (*Eriksen et al., 2015*; *Royle et al., 2005*). In the daffodil cichlid in particular, adult body size is a key determinant of lifetime reproductive success because it correlates positively with fecundity in females and with dominance acquisition in both sexes (*Heg et al., 2011*; *Wong and Balshine, 2011*).

## Discussion

Our experimental results show that chronic elevation of outgroup conflict negatively affected interclutch intervals and investment in egg quality; whilst there was a compensatory positive influence on parental care, an increased outgroup threat ultimately resulted in fewer offspring of smaller size surviving to 1 month of age. Our findings are in-line with previous correlational work showing that greater levels of intergroup threat were associated with reductions in chimpanzee reproductive rate and offspring survival (*Lemoine et al., 2020*), but contrast the observational studies of crested macaques and banded mongooses that found a positive correlation between intergroup conflict and foetal survival (*Kerhoas et al., 2014*; *Thompson et al., 2017*). Tests of causal links between

outgroup conflict and reproductive success have previously been lacking due to the inherent difficulties of carrying out long-term manipulations that allow cumulative fitness impacts to be assessed. We overcame this by using a model fish species that enables both extended manipulations in controlled conditions and the collection of behavioural and reproductive data (*Wong and Balshine, 2011*). In doing so, we demonstrated that chronic outgroup conflict likely negatively impacts the fitness of multiple generations: adults suffer reductions in current reproductive output, whilst the lower quality of surviving offspring means that they are potentially less likely to achieve dominance and thus direct reproductive success later in life (*Royle et al., 2005*; *Wong and Balshine, 2011*).

We uncovered outgroup-conflict effects in several aspects of reproductive investment and output that became stronger the longer the experimental treatment continued. These findings highlight that, to ascertain the full fitness consequences in social animals, it is important to consider both direct and indirect effects and to assess cumulative impacts in addition to those arising from single contests (*Morris-Drake et al., 2022*). Since outgroup conflict can be a potent social stressor, capable of stimulating acute and chronic stress responses (*Samuni et al., 2019*), stress is a likely mechanism underpinning the reproductive consequences seen. Studies examining natural stress responses (*Beldade et al., 2017*; *Creel et al., 2009*), and experimental manipulations of glucocorticoids (*Ensminger et al., 2018*; *Eriksen et al., 2015*; *McCormick, 1998*), have demonstrated how stress can affect adult reproductive measures in ways similar to those uncovered in our work. For instance, chronic anemone bleaching and predation risk led to increased levels of stress hormones and reduced reproductive rates in the anemonefish, *Amphiprion chrysopterus*, and in elk, *Cervus elaphus*, respectively (*Beldade et al., 2017*; *Creel et al., 2009*). Experimental increases in maternal cortisol levels negatively impacted egg protein content of Eastern fence lizards and larval length of the coral reef fish *Pomacentrus amboinensis* (*Ensminger et al., 2018*; *McCormick, 1998*). Higher rates of territorial defence have previously been associated with elevated cortisol levels in *N. pulcher* dominant females (*Culbert et al., 2021*), so treatment effects on maternal physiology could plausibly have contributed to the effects seen in our study. Likewise, early-life stressors can have long-lasting effects on offspring phenotypes and fitness (*Antunes and Taborsky, 2020*; *Jonsson and Jonsson, 2014*), so both direct and indirect effects may be important. Some of the fitness costs that we document here may also have been indirectly caused by outgroup conflict-mediated disruptions to within-group social relationships. Within-group aggression is promoted by the mere presence of neighbours in complex time-, rank-, and sex-dependent ways (*Hellmann and Hamilton, 2019*) and may be further dependent on the extent of interactions between residents and neighbours (*Hamilton and Heg, 2005*). Additionally, territorial intrusions can cause (at least) short-term changes to within-group social interactions (*Braga Goncalves and Radford, 2019*; *Bruintjes et al., 2016*). Cumulative effects of outgroup conflict are likely to have a wide influence, beyond reproductive success, and thus deserve further research attention moving forward (*Morris-Drake et al., 2022*).

There are obvious trade-offs in conducting captive versus field studies. In the context of cumulative outgroup-conflict consequences, long-term experimental manipulations are logistically and ethically challenging, which is why the few previous studies have reported only correlational data (*Kerhoas et al., 2014*; *Lemoine et al., 2020*; *Thompson et al., 2017*). Our captive experiments allowed for tight control of the environment and close monitoring of a variety of response measures, but consideration is needed of the ecological validity of the intrusion regime. Whilst intrusions in the wild might be, on average, shorter than our 10-min experimental presentations, there are several reasons to believe that outgroup conflict could be more severe in natural conditions. In the wild, *N. pulcher* groups are often surrounded by several adjacent groups (*Taborsky, 1984*), so that territories are intruded from any direction by neighbours and outsiders from further afield. Therefore, intrusions likely take place several times daily, requiring continuous vigilance and quick behavioural responses, risking potential physical injury and leaving residents more vulnerable to predation (*Balshine-Earn et al., 1998*; *Balshine et al., 2001*; *Hess et al., 2016*). Our setup also precluded individuals from injuring each other, the intruder from committing infanticide, sneaking a breeding attempt or taking-over part or all the territory, and individuals from having to trade-off foraging time against vigilance efforts (due to plentiful food availability), all in the absence of predation. Thus, relative to wild conditions, our laboratory intrusions were likely longer but less frequent and came from only one pair of neighbours, and posed relatively low threat to focal groups within an overall less challenging environment. Whilst we chronically increased the perception of outgroup conflict, allowing us to assess cumulative

indirect effects of outgroup conflict (beyond injury, death, and infanticide), our findings are potentially conservative. Ultimately, future field experiments are needed to confirm the causal impact of chronic outgroup conflict on reproductive success, but many challenges will need to be overcome first.

We demonstrate that chronically increased outgroup conflict can reduce reproductive success, even in the absence of physical fights with rivals that may also result in injury, death, or loss of territory or breeding position (*Goodall, 1986*; *Spong et al., 2008*; *Thompson et al., 2017*), or cause offspring death (*Dyble et al., 2019*; *Thompson et al., 2017*). Outgroup conflict can clearly, therefore, influence fitness in myriad ways, lending support to the theory that it is likely a powerful selective force in social evolution with respect to, for example, group dynamics, social structure, and cooperation (*Bowles, 2009*; *Gaston, 1978*; *Wrangham, 1980*). Given the widespread taxonomic occurrence of outgroup conflict, we advocate further experimental testing of what is arguably the most neglected aspect of sociality.

## Materials and methods

### Fish husbandry

We used a captive population of daffodil cichlids, *N. pulcher*, at the University of Bristol; work was approved by the University of Bristol Ethical Committee (University Investigator Number: UB/16/049 + UB/19/059). All fish groups were housed in 70 l tanks (width × length × height: 30 × 61 × 38 cm) that formed their territory (as in *Braga Goncalves et al., 2021*; *Braga Goncalves and Radford, 2019*). Each tank contained 2–3 cm of sand (Sansibar river sand), a 75 W heater (Eheim), a filter (Eheim Ecco pro 130), a thermometer (Eheim), two flowerpot halves (10 cm wide) that served as breeding shelters at the centre of the territory, an artificial plant and a small tube hanging close to the water surface to provide extra shelter for the subordinate in focal groups. Fish were fed twice daily: alternating between frozen brine shrimp, water fleas, prawns, mosquito larvae, mysid shrimp, bloodworms, cichlid diet, spirulina, copepods and krill in the mornings from Monday to Friday; and dry fish flakes in the evenings and weekends. Water temperature was maintained constant (mean ± SE, Experiment I: 26.9 ± 0.08°C, Experiment II: 26.7 ± 0.10°C) and room lights were set on a 13L:11D hour cycle (daylight from 7 AM to 8 PM). Water quality tests (pH, nitrates, nitrites, conductivity, and ammonia) and 10% water changes were performed weekly to maintain water quality levels.

### Experimental setup

In both experiments, we organised tanks in triplets (*N* = 10), positioned in a line with the short sides about 0.5 cm apart, so that neighbours could always see each other. In each triplet, the central tank housed a breeding pair that was a common neighbour to the two focal groups (each comprising a breeding pair and an adult helper) on either side. Groups of three, although at the low range of natural group sizes for this species, are common in nature (*Balshine et al., 2001*) and frequently used in laboratory studies (*Braga Goncalves and Radford, 2019*; *Fischer et al., 2017*; *Hamilton and Ligocki, 2012*; *Mileva et al., 2011*; *Zöttl et al., 2013*). We formed groups separately (with different fish used for the two experiments) using standard procedures (*Braga Goncalves and Radford, 2019*) approximately 2 weeks prior to the start of each experiment. In Experiment I, the 20 focal groups included 12 with female helpers and 8 with male helpers; within a triplet, experimental groups had same-sex helpers. In Experiment II, the 20 focal groups all had female helpers. Female helpers were favoured because although all subordinates face increasing risks of eviction as they grow (*Taborsky, 1985*), males are more likely to parasitise reproductive events resulting in eviction from the group (*Dierkes et al., 1999*) rendering groups less stable, and group size has been shown to affect egg parameters (*Taborsky et al., 2007*). To minimise same-sex aggression and to aid individual identification, we ensured that each dominant was at least 5 mm larger than the same-sex subordinate in the focal group (mean ± SE size difference, Experiment I: 10.7 ± 1.0 mm; Experiment II: 22.4 ± 1.3 mm). To minimise potential size-difference effects between treatments, and control for the influence of female size on reproductive measures (*Heg et al., 2011*), we size-matched (in standard length) breeders in all tanks in a triplet to each other. At the start of Experiment I, focal-group dominant males were 55.2 ± 1.3 mm, dominant females were 53.4 ± 0.9 mm, subordinate males were 41.8 ± 0.9 mm and subordinate females were 44.3 ± 1.3 mm long, while males were 59.3 ± 2.6 mm and females were 54.8 ± 1.6 mm long in the middle-tank breeding pairs. At the start of Experiment II, focal-group dominant

males were 73.1 ± 1.3 mm, dominant females were 64.4 ± 0.8 mm and subordinate females were 41.8 ± 1.0 mm long, while males were 71.4 ± 2.7 mm and females were 63.6 ± 1.8 mm long in the middle-tank breeding pairs.

For each triplet, we randomly allocated (by flip of a coin) the side tanks to one of two experimental treatments, Control and Intruded; the central tank provided the intruders used in the Intruded treatment. Experiment I spanned 13 weeks to allow for multiple spawning bouts; we destroyed clutches produced before groups had experienced at least three intrusions (Control = 1 clutch, Intruded = 2 clutches) as groups would likely not have been significantly affected by treatments within such a short timeframe. In Experiment II, following eight intrusions during the first 13 days, the first clutch produced by each group marked the end of treatment for that group; 2 weeks represents approximately half the time of a regular breeding cycle (*Desjardins et al., 2011*; *Jindal et al., 2017*). Eleven clutches were destroyed during the initial 13 days of treatment (Control = 8 clutches, Intruded = 3 clutches). Previous destruction of a clutch had no significant impact on egg dry weight ($\chi^2_1$ = 0.21, p = 0.650), egg volume ($\chi^2_1$ = 0.05, p = 0.833), lipid content ($\chi^2_1$ = 0.16, p = 0.692), or protein content ($\chi^2_1$ = 0.89, p = 0.345). Experiment II was run for 11 weeks to maximise the number of focal groups that produced a clutch.

## Simulated territorial intrusions

We selected the side of the focal Intruded tank where each intrusion took place (near or far from the neighbour tank) and the identity of the intruder (male or female neighbour) pseudo-randomly: by flip of a coin, but no more than four of the same side or sex in a row. In the wild, dominant individuals of both sexes undertake regular forays to nearby territories (*Jungwirth et al., 2015*) and may, thus, return to their own territories from any direction. At the start of an intrusion (or equivalent in the Control tanks), we slid down one transparent and one opaque flexible partition (0.75 mm white ViPrint) through single-channel PVC tracks glued to the long walls, 8 cm from the tank edge, creating a side compartment at the edge of the territory of the focal group (*Braga Goncalves et al., 2021*; *Braga Goncalves and Radford, 2019*). Then, we netted out the pre-selected neighbour and placed it in the side compartment of the focal tank, obscured from view of the resident group for a 5-min settling period. Brief handling experiences do not affect behaviour adversely in this species (*Braga Goncalves and Radford, 2019*; *Mileva et al., 2009*). After the settling period, we removed the opaque partition in the focal tank to reveal the intruder to the resident group for 10 min, a trial duration that falls comfortably with the range (5–20 min) often used in laboratory and field studies in this species (*Desjardins et al., 2008*; *Jungwirth et al., 2015*; *Reyes-Contreras et al., 2019*). At the end of the intrusion period, we replaced the opaque partition in the focal tank and placed another opaque partition between the focal and neighbour tanks, netted the intruder and transferred it back to the neighbour tank out of sight of the focal group. Concurrently, in the matching Control group, we conducted the same sequence of placement and removal of opaque and transparent partitions as in the Intruded tank, but in the absence of an intruder. To minimise the impact of human presence on the behaviour of the experimental subjects, the experimenter hid behind a curtain during the period that the intruder was visible to the resident group.

In Experiment I, we filmed (Sony Handycam HDR-XR520) two simulated intrusions, one at the start (week 1) and another towards the end (week 11) of the study, to assess how focal groups responded to the presence of a neighbour in their territory, and how their response changed over time. For consistency, all filmed intrusions had the female neighbour as intruder placed on the side of the tank closest to the neighbour tank. Similarly, we filmed the behavioural responses of the Control groups to the presence of the transparent partition in their territory. Subordinate sample size was smaller (*N* = 12) at the end of the study due to deaths unrelated to the experiment (fish jumped out of tank) and evictions (Control = 4, Intruded = 4). Following previously established protocols for this species (*Reddon et al., 2015*; *Sopinka et al., 2009*), we recorded using JWatcher (v1.0; Macquarie University, Sydney, Australia) the frequencies of aggressive behaviours, including attacks (rams and bites) and aggressive displays (aggressive postures, frontal displays, and fast approaches), directed at the intruder or the transparent partition by each group member. From the Intruded videos, we also recorded intruder responsiveness towards the focal group, by assessing proportion of time active and facing the focal group, as in *Braga Goncalves and Radford, 2019*.

## Reproductive rate

To assess the impact of outgroup conflict on the likelihood of spawning, latency to spawn, number of clutches produced, and mean inter-clutch interval (Experiment I), we scanned all focal tanks every day for new clutches.

## Investment in eggs

To evaluate the impact of outgroup conflict on clutch size, we photographed all clutches in Experiment I ($N = 34$) the day after they were laid and counted the numbers of eggs using ImageJ (version 1.46r, National Institutes of Health, USA). Hatching success could only be assessed in a subset of clutches ($N = 21$) because the hatchlings of several clutches took cover on the sand or on the plant and could not be counted reliably.

To evaluate the impact of outgroup conflict on egg size and nutritional content, we separated from each clutch in Experiment II: (1) 10–12 eggs for assessment of egg size; (2) two samples of 6–11 eggs for lipid analysis; and (3) two samples of 5 eggs for protein analysis. Two clutches (1 Control, 1 Intruded) were laid over 2 days; we collected two samples for egg-size assessment from these clutches. Samples (2) and (3) were stored at −20°C until extraction.

Using the egg-size sample, we measured the length and width of each egg with a stereo microscope (×2 magnification) and a graticule. We then calculated the effective diameter of each egg (i.e. the diameter if it were perfectly round) as the cube root of the length multiplied by the square of the width, and used the effective diameter to calculate the volume assuming a spherical shape (as per *Coleman, 1984*). We used all egg volumes from each sample in the analysis. After the eggs were individually measured, all eggs from the same clutch were placed together in a petri dish with wax paper and dried in a heating cupboard at 70°C for 48 hr, weighed twice (ME5, Sartorius, Göttingen, Germany), returned to the heating cupboard for another 24 hr and weighed twice again; all four measurements were used in the analysis.

To assess the total lipid content of eggs, we used the colorimetric sulfo-phosphato-vanillin method for microquantities (*Alqurashi et al., 2019*). Briefly, we dried the samples at 70°C for 24 hr and weighed them twice to the nearest microgram (CPA26P, Sartorius, Göttingen, Germany). We transferred the samples into 15 ml round-bottom glass test tubes (16 × 150 mm) and crushed them with a glass rod before adding 10 ml of chloroform–methanol solution (1:1, vol/vol). We extracted 0.5 ml of supernatant from each sample into new test tubes and placed them in a dry bath (LSE single block digital; Corning Ltd, Barry, UK) at 100°C for 15 min to evaporate the solvent. After, we added 0.2 ml of sulphuric acid to the samples and placed them in the dry bath for a further 10 min. Once the samples had cooled to room temperature, we added 4.8 ml of vanillin- ortho-phosphoric acid 85% reagent (1.2 g/l), thoroughly mixed the samples for 30 s with a vortex mixer (Bibby Scientific, Stone, UK), and transferred 1 ml of the solution into 1 ml polystyrene semi micro cuvettes. Using a spectrophotometer (WPA Biowave UV/Vis; Biochrom Ltd, Cambridge, UK) at 525 nm, calibrated with a vanillin/phosphoric acid only blank, we took three absorbance readings from each sample and calculated their mean. The mean sample absorbance values were plotted against a standard curve to extrapolate their total lipid content (µg). The standard curve was prepared using eight serial dilutions of analytical soybean oil solution (0.917 g/ml, Sigma Aldrich) in methanol:chloroform (1:1).

To assess the total protein content of eggs, we used the Bradford method (*Bradford, 1976*) following specifications by *Alasmari and Wall, 2020*. The egg samples were transferred into 15 ml round-bottom borosilicate glass test tubes (16 × 150 mm) and crushed with a glass rod. We added 0.5 ml of phosphate buffer (100 mM of monopotassium phosphate [$KH_2PO_4$], 1 mM of ethylenediaminetetraacetic acid, and 1 mM of dithiothreitol dissolved in distilled water), mixed with an aqueous solution of potassium phosphate dibasic ($K_2HPO_4$) to achieve pH = 7.4 to dissolve the sample, followed by another 0.5 ml of buffer to clean the glass rod. After, we thoroughly mixed the samples for 30 s with a vortex mixer (Bibby Scientific, Stone, UK), we transferred 0.1 ml into new tubes, added 2.9 ml of buffer, and vortexed the new solutions for another 30 s to perform 1:30 dilutions. From these dilutions, we transferred 1 ml of the solution into a new tube, added 1 ml of Bradford reagent, and vortexed the resulting solution for 1 min. After waiting 5 min at room temperature, we transferred 1 ml of the solution into 1 ml polystyrene semi micro cuvettes and read the absorbance values on a spectrophotometer at 595 nm, calibrated with a buffer and Bradford reagent-only blank. Each sample was measured three times and the mean value was plotted against a standard curve to extrapolate

the total protein content in the sample. The standard curve was prepared using eight serial dilutions of bovine serum albumin (1 mg/ml, Sigma) between 0 and 50 µl diluted in buffer to a total volume of 1 ml. Protein content estimates were multiplied by 30 to express protein content per egg (µg).

### Investment in parental care

To assess the impact of outgroup conflict on parental care, we filmed (Sony Handycam HDR-XR520) each focal group in Experiment I for 10 min on the morning after a clutch was laid, before they experienced an intrusion that day. Parental care at the egg stage, consisting of fanning and cleaning the eggs, aids embryonic development and survival (*Taborsky, 1984*). From the videos, we recorded the number of clutch visits (visits to within a body length of the clutch) and parental-care behaviours (egg-cleaning and clutch-fanning combined) displayed by all group members, as well as the total time spent in clutch visits and in parental-care behaviour.

### Reproductive output

To assess the impact of outgroup conflict on offspring survival, we visually counted the number of surviving fry on the day that they reached 1-month post-hatching in Experiment I (*N* = 32 broods). To count offspring numbers in larger clutches reliably, we temporarily divided the tanks into three sections using transparent partitions and counted the number of offspring in each section separately. All counts were done twice to confirm totals; where values differed, we counted young a third time and either took the confirmed number of young or calculated the mean number from the three counts.

To assess the impacts of outgroup conflict on offspring activity levels and response to a sudden stimulus, we tested offspring at 1-month post-hatching. At this age, offspring actively explore the territory (i.e. swim around the tank) and, when they perceive a conflict, they sink to the substrate and remain immobile to blend with the sandy background (*Taborsky, 1984*; *Watve and Taborsky, 2019*). A sample of 5–10 offspring (*N* = 21 clutches) was transferred from their home tank to a test container (20 × 20 × 10 cm), filled with 3 l of water from their home tank, and left to settle for 30 min before the commencement of a trial. Each trial was filmed (Sony Handycam HDR-XR520) from above. After an initial 5 min undisturbed, pre-stimulus period, we released a small glass marble in a 60-cm plastic tube placed at a right angle relative to and touching the side of the container, so that the marble produced sudden vibrations and noise as it hit the container. We then filmed offspring behaviour for a 5-min post-stimulus period. We calculated mean offspring activity level pre-stimulus by observing 3 s of film every 20 s and recording how many offspring were actively swimming during that period. We also took screenshots every 20 s during the 5 min pre-stimulus period, from which we measured the mean nearest-neighbour distance of the offspring. From the post-stimulus period, we recorded the latency for all individuals to become immobile in response to the stimulus and the latency for the first offspring to become active again.

After the startle-stimulus trials, we euthanised the offspring with an overdose of tricaine methanesulfonate (MS222, 12 ml/ 100 ml tank water) and stored them in a 30% ethanol freshwater solution, shown to be adequate for simultaneously preserving body tissues and minimising shrinkage (*Gagliano et al., 2006*). We measured offspring standard length (from the tip of the snout to the end of the caudal peduncle, ±0.01 mm) using Leica Application Suite (version 4.4.0 [Build:454], Leica Microsystems Limited, Switzerland) connected to a camera (Greenough Stereozoom ×0.8 manual) mounted on a stereo microscope (EZ4 HD, eyepiece ×10/21B). Immediately after measuring the offspring, we dried them for 36 hr at 70°C before weighing them individually on a Mettler scale (AE260, Delta Range, ±0.1 mg). We then used the individual measurements to calculate mean offspring standard length and dry weight per clutch.

### Statistical analyses

All statistical analyses were conducted using RStudio (version 1.2.5033, *RStudio, 2020*). In Experiment I, we used paired *t*-tests or Wilcoxon signed-rank tests (depending on whether datasets met the assumptions of parametric testing or not, respectively) to assess intruder behaviour and defensive contributions between treatments and within individuals at the start and end of the study; the latter were adjusted using the Bonferroni–Holm sequential method (*Dobson and Barnett, 2008*; *Holm, 1979*). We used a McNemar test and a Wilcoxon rank sum test to assess treatment differences in the likelihood of spawning and in the number of clutches produced, respectively.

For all other analyses, we used mixed-effects models. We visually assessed model assumptions and performance using the package 'performance' (*Lüdecke et al., 2021*). Where appropriate, we fitted LMMs with Gaussian error distributions to the raw data (package 'lme4' version 1.1-21, 99). Where datasets did not fit the assumptions of linear models, we fitted appropriate GLMMs where possible: a GLMM with negative binomial error distribution and log-link function (package 'glmmTMB' version 1.1.2.3, *Brooks et al., 2017*) for the analyses of the number of nest visits and clutch caring events; and a GLMM with Gaussian error distribution and log-link function (package 'lme4' version 1.1-21, *Bates et al., 2015*) for the analyses of time spent on clutch care. We $\log_{10}$-transformed latency to spawn to conform with linearity assumptions. The analysis of number of offspring to survive to 1 month of age included clutch size at laying as an offset. All models included tank-triplet identity and, where relevant, focal-group identity nested within tank-triplet identity as random factors to control for the shared neighbours within each triplet and for repeated observations for groups, respectively. Our main fixed factors of interest were treatment (Intruded and Control) and its interaction with treatment duration (covariate); we controlled for the effects of several covariates in different models as appropriate (full details of factors used in each analysis are provided in the *Supplementary file 1*).

In each model, we assessed term significance by comparing a model with and without the specific term using likelihood ratio tests (chi-square tests using R function 'anova'; *Dobson and Barnett, 2008*). Non-significant interaction terms were removed to enable us to assess the effects of the main factors independently (*Dobson and Barnett, 2008*; *Engqvist, 2005*); the resulting final models contained all mains factors and significant interaction terms. The effects of significant interactions between treatment and treatment duration were teased apart by analysing the effect of treatment duration on each treatment separately. In the main text, we provide significant PEs and associated 95% CIs or treatment effect sizes of the main factors of interest (i.e. treatment and its interaction with treatment duration); all PEs, CIs, and associated statistical outputs are provided in the Supplementary files.

## Acknowledgements

We thank Richard Wall for access to his laboratory facilities, Shatha Alqurashi and Saeed Alasmari for assistance in developing protein and lipid extraction protocols, Martin Aveling for the artwork in *Figure 1*, Barbara Taborsky and Michael Taborsky for detailed discussions about this project and the study species across the years, and Stephanie King, Ben Ashton, and Patrick Kennedy for comments on the manuscript. This work was supported by a European Research Council Consolidator Grant (project no. 682253) awarded to ANR. Funding: This work was supported by a European Research Council Consolidator Grant (project no. 682253) awarded to ANR.

## Additional information

### Funding

| Funder | Grant reference number | Author |
| --- | --- | --- |
| European Research Council | 682253 | Andrew N Radford |

The funders had no role in study design, data collection, and interpretation, or the decision to submit the work for publication.

### Author contributions

Ines Braga Goncalves, Conceptualization, Data curation, Formal analysis, Validation, Investigation, Visualization, Methodology, Writing – original draft, Project administration, Writing – review and editing; Andrew N Radford, Conceptualization, Funding acquisition, Project administration, Writing – review and editing

### Author ORCIDs

Ines Braga Goncalves http://orcid.org/0000-0003-0659-9029
Andrew N Radford http://orcid.org/0000-0001-5470-3463

### Ethics

Our work was approved by the University of Bristol Ethical Committee (University Investigator Number: UB/16/049 + UB/19/059).

### Decision letter and Author response

Decision letter https://doi.org/10.7554/eLife.72567.sa1
Author response https://doi.org/10.7554/eLife.72567.sa2

---

## Additional files

### Supplementary files

• Supplementary file 1. Statistical summary of linear mixed models testing the effect of chronic outgroup conflict (Intruded vs. Control, Experiment I) on breeding timings. Effect of outgroup conflict on (a) latency to spawn (days) and (b) inter-clutch interval (days); latency to spawn was $\log_{10}$-transformed. Female size relates to dominant female standard length at the start of the study. Tank-triplet and/or group identity nested within tank-triplet were fitted as random intercepts (with variances shown). The reference level for Treatment was Control. Each table section displays the final model, with removed non-significant interactions below.

• Supplementary file 2. Statistical summary of linear mixed models testing the effect of chronic outgroup conflict (Intruded vs. Control) on clutch size. Effect of outgroup conflict on clutch when females could spawn repeatedly throughout the study (Experiment I). Female size relates to dominant female standard length measurement made closest in time to the production of each clutch (start or end of study). Tank-triplet and group identity nested within tank-triplet were fitted as random intercepts (with variances shown). The reference level for Treatment was Control. Table displays the final model, with removed non-significant interactions below.

• Supplementary file 3. Statistical summary of linear mixed models testing the effect of chronic outgroup conflict (Intruded vs. Control, Experiment II) on mean morphological and physiological egg characters. Effect of outgroup conflict on egg (a) volume ($mm^3$), (b) dry weight (mg), (c) lipid content (μg), and (d) protein content (μg). Female size relates to dominant female standard length at the start of the study. Tank-triplet and group identity nested within tank-triplet were fitted as random intercepts (with variances shown). The reference level for Treatment was Control. Each table section displays the final model, with removed non-significant interactions below. For fixed effects included in significant interactions, only parameter estimates are shown.

• Supplementary file 4. Statistical summary of generalised linear mixed models (GLMMs) testing the effect of chronic outgroup conflict (Intruded vs. Control, Experiment I) on the number of parental-care behaviours performed during a 10-min period. Effect of outgroup conflict on (a) clutch visits and (b) caring (egg-cleaning and fanning) events; both analysed using negative binomial GLMMs with a 'log' link function. Tank-triplet and group identity nested within tank-triplet were fitted as random intercepts (with variances shown). The reference level for Treatment was Control. Each table section displays the final model, with removed non-significant interactions below. For fixed effects included in significant interactions, only parameter estimates are shown.

• Supplementary file 5. Statistical summary of mixed models testing the effect of chronic outgroup conflict (Intruded vs. Control, Experiment I) on time spent on parental-care behaviour during a 10-min period. Effect of outgroup conflict on time spent on (a) clutch visits (linear mixed model, LMM) and (b) caring (egg-cleaning and fanning) events (generalised linear mixed model [GLMM] with 'gaussian' family and 'log' link function). Tank-triplet and group identity nested within tank-triplet were fitted as random intercepts (with variances shown). The reference level for Treatment was Control. Each table section displays the final model, with removed non-significant interactions below. For fixed effects included in significant interactions, only parameter estimates are shown.

• Supplementary file 6. Statistical summary of a linear mixed model testing the effect of chronic outgroup conflict (Intruded vs. Control, Experiment I) on hatching success (%). Tank-triplet and group identity nested within tank-triplet were fitted as random intercepts (with variances shown). The reference level for Treatment was Control. Table shows the final model with removed non-significant interactions below.

• Supplementary file 7. Statistical summary of a linear mixed model testing the effect of chronic outgroup conflict (Intruded vs. Control, Experiment I) on offspring survival to 1-month post-hatching. Tank-triplet and group identity nested within tank-triplet were fitted as random intercepts (with variances shown); clutch size at laying was included as an offset. The reference level for Treatment

was Control. Table displays the final model. For fixed effects included in the significant interaction, only parameter estimates are shown.

• Supplementary file 8. Statistical summary of linear mixed models testing the effect of chronic outgroup conflict (Intruded vs. Control, Experiment I) on offspring behaviour. Effect of outgroup conflict on (a) mean pre-stimulus activity (%), (b) mean pre-stimulus nearest-neighbour distance (cm), (c) mean latency to freeze (s) post-stimulus, and (d) latency (s) to become active post-stimulus. Tank-triplet and group identity nested within tank-triplet were fitted as random intercepts (with variances shown). The reference level for Treatment was Control. Each table section displays the final model, with removed non-significant interactions below. For fixed effects included in significant interactions, only parameter estimates are shown.

• Supplementary file 9. Statistical summary of linear mixed models testing the effect of chronic outgroup conflict (Intruded vs. Control, Experiment I) on offspring size. Effect of outgroup size on (a) mean standard length (mm) and (b) mean dry weight (mg). Tank-triplet and group identity nested within tank-triplet were fitted as random intercepts (with variances shown). The reference level for Treatment was Control. Each table section displays the final model, with removed non-significant interactions below.

• Transparent reporting form

• Source data 1. Source data for all statistical analyses (data for each separate analysis are provided in separate labelled worksheets).

### Data availability
Source data files have been uploaded for each of our figures and all other datasets are uploaded as Supplementary files.

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
