## [Editor Report]

This paper experimentally investigates the fitness consequences of intergroup conflict in social fish. It finds that groups that face frequent territorial intrusion suffer costs in terms of both fertility and number of surviving offspring, despite behavioral compensation through increased parental care. These results provide clear and compelling evidence that intergroup conflict leads to lower fitness, and are therefore of substantial interest for understanding social evolution, where the importance of between-group competition has long been debated.

---

## [Decision Letter]

**Decision letter after peer review:**

Thank you for submitting your article "Experimental evidence that chronic outgroup conflict reduces reproductive success in a cooperatively breeding fish" for consideration by *eLife*. Your article has been reviewed by 3 peer reviewers, one of whom is a member of our Board of Reviewing Editors, and the evaluation has been overseen by Detlef Weigel as the Senior Editor. The following individual involved in review of your submission has agreed to reveal their identity: Mark Dyble (Reviewer #3).

Essential revisions:

1) Eliminate stepwise model selection procedure to minimize the risk of false positives. Report either betas and p-values from the full model, or via removal of the effect of interest from the full model, followed by comparison of model fit (e.g., using likelihood ratio tests).

2) Explain the experimental paradigm early in the manuscript to facilitate interpretation by readers. Key points include the fact that the intrusion paradigm involved no actual physical contact, and that in Experiment 2, all clutches prior to Day 13 were destroyed. Clearly motivate the analyses that are later presented in the Results section by reference to description of the experimental paradigm.

3) Intergroup conflict effects on the number of surviving offspring, the closest proxy to fitness used here, are one of the most important results of the paper. If possible, present a straightforward comparison of this outcome for the treatment versus control groups across the duration of the experiment, along with a simple analysis of the magnitude of this difference (e.g., using a paired t-test). Clarify the degree to which conflict effects on surviving offspring are magnified over time (e.g., in terms of the % of additional surviving offspring in the control group, across a given number of days since the start of the experiment).

4) Pay close attention to the reviewer recommendations on statistical analysis below, especially consistency in declaring effects "significant"/"near significant" and reporting of "near-significant" effects.

*Reviewer #2 (Recommendations for the authors):*

– To address my concern about stepwise model reduction, the authors should report the significance of each effect by comparing the fit of a full model to that of a model without that effect. A second option would be to remove only non-significant interaction terms, if the goal is mainly to investigate the fixed effects (Enqvist 2005 An. Beh., but see Shielzeth 2010 Meth. Ecol. Evo.). I often take the second approach in my own work, but am realizing that the first approach may be more appropriate.

– To address my concern about a lack of clear hypotheses, the authors could outline why each specific metric was measured and why each relates to fitness early on (e.g., at the beginning of each paragraph or at the end of the Introduction).

Comments below are line-by-line

Title: "outgroup conflict" might be a less widely-used term than "intergroup conflict" in this literature. For example, the titles of most of the first 10 references in this paper use "intergroup" instead of "outgroup". Consider using "intergroup" instead of "outgroup".

– Line 43: consider citing Green et al., 2020, which is a review of intergroup contest studies and how they focus on factors like the determinants of intergroup contest success. Green PA, Briffa M, Cant MA. 2020. Assessment during intergroup contests Trends Ecol Evol. 36(2):139-150.

– Line 47: consider citing Preston et al., 2020, which looked at short-term and longer-term (3 days) changes in grooming and aggressive behavior within groups after simulated intergroup conflict. Preston EFR, Thompson FJ, Ellis S, Kyambulima S, Croft DP, Cant MA. 2020. Network‐level consequences of outgroup threats in banded mongooses: Grooming and aggression between the sexes. J Anim Ecol.

– Lines 54-55: I had trouble telling whether there was a timescale difference in these three studies. Did the studies truly differ in a focus on "foetal survival" (first two) versus "infant survival" (third)?

– Lines 56-57: It would be useful to have some more detail about why we really need these experiments. What is missing from those observational studies that an experimental study could really help us learn? Put another way, can you give more details on the importance of this "causal link"?

– Lines 88-96: It's unclear to me from these results and Figure 2 why a Wilcoxon test versus a t-test was used for certain tests. My assumption is that this relates to the normality of the data distributions. Perhaps a simpler way to get at the same question would be to build a linear mixed model predicting # of defensive behaviors from treatment, time, and individual type (dominant male/female, subordinate)?

– Results in general: I'm a bit confused as to why significant effects are reported for certain interaction terms, and then also for each level of the interaction. For example, lines 180-184 detail the significant interaction (and associated P-value from a Chi-squared test), as well as reporting t-values and P-values from each level of the interaction (e.g., treatment duration effects within each treatment). I believe just reporting the P-value from the Chi-squared test for the interaction term shows that the interaction is significant, which should then be followed by simpler reporting of the estimate for each level of the interaction. It seems unnecessary to test whether each level of the interaction is also significant.

– Results in general: There are several examples where the authors report "trending" or nearly-significant results that match their predictions (e.g., lines 161-163; 193-194; 216-217; 266). However, there are also instances in the reporting of the statistical models in which similarly trending factors were removed from statistical models (e.g., Table S2; S4; S6; S8; S9). I believe some of these removed values were reported in the results (e.g., line 266 matches to Table S9). However, it is confusing to understand why certain terms were removed from the statistical models, especially if they are reported in the Results (i.e. we can assume the authors consider these are important). See above for comments regarding statistical analyses.

– Figure 3 and all scatterplots: are these lines showing best fit from the LMM, i.e. the estimates from the LMM? Or are they plotting simply using a function like geom_smooth in ggplot? Showing the actual predicted estimates from the model on the scatterplot would give the reader a better idea of both the raw data (from the points) and the model estimates (from the lines and associated ribbons of confidence intervals). There are also methods to do this for boxplots. All of these methods can be accomplished using the "effects" package in R.

– Lines 166-168: This is a bit confusing, because I thought the protein and lipid analyses were only conducted in Experiment II on eggs from the first clutch? So, how can conclusions about protein/lipids in the eggs from later clutches be made if only the first clutch was analyzed? Or, is this because clutches were analyzed across groups, i.e., several clutches for a given focal group were not analyzed, but across the entire experiment the first clutch from each group gave a time effect?

– Figure 4B: There appears to be an outlier with an incredibly high number of cleaning events at approximately 60 days treatment duration in the Intruded group. It would be useful to know if there are any biological differences between this observation and the others. Also, if this is a statistical outlier, it would be useful to know whether removing it affects the results. While I don't suggest the removal of statistical outliers simply because they're outliers (i.e., there should be a biological reason for data to be removed), it's important to take not of any outlier values like this.

– Lines 219-224: I might be being dense here, but I'm unsure of what this sentence is saying. How is the relationship (or lack thereof) between the number of hatched fry and the number of offspring reaching 1 month of age not offspring survival? It seems from Table S7 that there was a difference between offspring survival to 1 month and survival from hatching to one month. I guess I'm having trouble understanding why we need both of these measures.

– Lines 257-259: Could another interpretation of the return to activity data be regarding boldness? That is, offspring from Intruded groups were bolder later on in the experiment?

– Lines 277-280: However, your results are counter to those of other studies you described in the Introduction, which found no or positive effects of intergroup conflict on reproduction and survival. Consider re-reporting that work here, as well, for consistency and clarity.

– Paragraph lines 311-340: I'm a bit confused as to the point of this paragraph. I understand the authors may want to compare the pros and cons of their experimental approach to a more field-based, correlational approach. However, the result of this comparison mainly convinces me that the experiment under-stressed the study animals, as compared to what they might experience in the wild. Further, the reporting of the lack of impacts on adult body mass, egg mass, etc… essentially make the point that the treatment did not affect groups much. This is counter to the rest of the paper's point, showing that groups were significantly affected. Do the authors have any hypotheses as to why they found specific results for the variables they did (e.g., egg protein, but not egg mass)? Do the authors have any thoughts or data on the relative reproductive success of these experimental groups compared to groups in the wild? I.e., are these experiments actually replicating stressors in the wild?

Methods line-by-line

– Lines 408-411: If one of the points of the present experiment was understanding the effects of short- versus long-term intergroup conflict, why destroy clutches produced before three intrusions? Wouldn't these give valuable information on short-term effects (or the lack thereof)?

– Line 549: was the number of offspring in the experiment considered as a factor predicting responses? In Table S8, I believe "number of offspring" is used as a fixed effect, but I was unclear as to whether that was number of offspring produced, or number used in the startle experiment. One might expect that the experimentally-designed groups with 10 offspring were different from those with 5.

– Lines 550-551: Was there any sand in the bottom of this startle tank? I would imagine that offspring sinking responses, if meant to blend into the substrate, would be different if there were no substrate to blend in to.

– Lines 585-589: Stepwise model reduction has come under heavy criticism in the past 15 years. See, for example, Forstmeier and Schielzeth (2011), Mundry and Nunn (2009), and Harrison et al., (2018). This process can greatly increase the likelihood of false-positive results, and can inflate parameter estimates. From my knowledge, removal of non-significant interaction terms is OK, if one is interested in achieving accurate parameter estimates for main effects. However, it is rarely advised to remove main effects. Ideally, the authors would report significance tests from the full model, or use information-theoretic approaches. However, the authors could consider removing only non-significant interaction terms, as a means of getting better estimates for the main effects (Engqvist 2005).

Forstmeier W, Schielzeth H. 2011. Cryptic multiple hypotheses testing in linear models: Overestimated effect sizes and the winner's curse. Behav Ecol Sociobiol. 65(1):47-55.

Mundry R, Nunn CL. 2009. Stepwise model fitting and statistical inference: Turning noise into signal pollution. Am Nat. 173(1):119-123.

Harrison XA, Donaldson L, Correa-Cano ME, Evans J, Fisher DN, Goodwin CED, Robinson BS, Hodgson DJ, Inger R. 2018. A brief introduction to mixed effects modelling and multi-model inference in ecology. PeerJ. 6:e4794.

Engqvist L. 2005. The mistreatment of covariate interaction terms in linear model analyses of behavioural and evolutionary ecology studies. Anim Behav. 70(4):967-971.

– Lines 603-607: I would appreciate learning more about this approach from the authors. It's very unclear to me why you wouldn't simply fit an appropriate response variable distribution in the initial model fitting step. For example, why not just use a Poisson distribution for the number of caring events response variable? This reads as though all response variables were forced into a Gaussian distribution. Other distributions are appropriate for other types of data, and do not require this step. I might be misunderstanding, however.

– Lines 609-612: I'm unclear as to why these clutches with zero hatching success were removed. The reasons these clutches had zero hatching success seem incredibly relevant to the study: if a clutch was eaten it seems evidence of cannibalism by the focal group (see lines 227-230 for a statement relevant to cannibalism in this species). If the clutch had a fungal infection, it would seem that the focal group's lack of care could have resulted in an infection.

*Reviewer #3 (Recommendations for the authors):*

My recommendations are superficial. My main comment is that I was frustrated by the cursory description of the experimental set up prior to the results. I know the *eLife* format is for the methods to go at the end of the manuscript but I really do think that at the very least the authors ought to explain how the simulated intrusions were enacted. The results didn't really come in to focus for me until after I had read the methods.

Line comments:

L70: on natural behaviour in the lab: have multiple groups been kept in the same tank and formed territories? If so, would be worth mentioning.

L84: This is where I would like to know how these territorial intrusions were simulated.

L100: In general, the paper is clearly written. However, I found this paragraph confusing and had to re-read several times. The description of the experimental set-up in the methods, in contrast, reads very clearly.

L126: '40%'. Can you give a 95% CI here?

L423-441: This is the critical detail on how the incursions were set up that I would have liked to see in the main text.

[Editors’ note: further revisions were suggested prior to acceptance, as described below.]

Thank you for resubmitting your work entitled "Experimental evidence that chronic outgroup conflict reduces reproductive success in a cooperatively breeding fish" for further consideration by *eLife*. Your revised article has been evaluated by Detlef Weigel (Senior Editor) and a Reviewing Editor.

The manuscript is much improved and close to publication-ready. However, the reviewers offered a few suggestions that may be helpful in communicating with your audience, so we are returning the draft to you to address these comments at your discretion.

1. Addressing Reviewer 2's notes about consistent reporting of significant parameter estimates.

2. Clarifying the set of behaviors coded as "defensive action".

*Reviewer #2 (Recommendations for the authors):*

Overall, I think the authors have done a great job of addressing my and the other reviewers' concerns. I am glad that the statistical suggestions have helped simplify the message, which is still an engaging and exciting one. I also greatly appreciate the laying out of predictions and the details of the methods early on. This will hopefully help readers get oriented. I have one main point, which I lay out first, and then a few line-by-line comments. I don't believe any of these comments are likely to change the main message of the manuscript.

Main comment:

On line ~680-681 in the Methods, the authors say they "provide significant parameter estimates…in the main text." However, looking at the Supplemental Tables in comparison to what is reported in the main text, only some significant effects are reported in the main text, not all. I'm curious as to why this is. For example, in the analysis of inter-clutch interval (Supp Table 1), female size had a highly significant effect (P = 0.017) but only the effect of treatment was discussed in the Results (lines 180-181). Similarly, in the analysis of egg investment, only the near-significant effect of the interaction between treatment and treatment duration was discussed (lines 195-197), but not the significant fixed effect of treatment. Also, an effect of clutch size on protein content and an effect of treatment x female size on protein content (Supp Table 3) were unreported in the Results. If these significant, but unreported, effects are not of interest, then one might ask why they were included in the model in the first place (aside from where necessary, like the treatment main effect when there is an interaction of interest). Or, are these omissions due to space constraints? There is likely a simple way to include a fuller reporting of these results. E.g., around line 195 the authors could report that "…there was a significant effect of treatment duration on clutch size (STATS HERE). This was likely driven by the interaction between treatment and treatment duration, in which Intruded females…." Or, an alternative would be in the Methods to state clearly that the authors only report selected results relevant to their hypotheses, but that some significant findings are reported only in the Supplementary Tables.

Line by line:

Lines 103 and 107: For the other predictions the authors make directional predictions, e.g. a negative effect. Can they specify the direction of the effect here? Or, are there no directional predictions?

Lines 132-151: These details are incredibly helpful! Thanks for including them.

Figure 3D: There are two red points at the end of treatment duration, right in the path of the line for the control (blue) effect. Are these points miscolored? It could just be an incredible chance that the line for control goes right through these intruded points, but I would also expect the 95% CI ribbon for intruded to be wider if these points are from the intruded treatment.

Supp Table 4A, Figure 4A, and lines 656-663: I could be wrong, but I would imagine that the number of clutch visits analysis should also be done using negative binomial error distribution, or a Poisson distribution. It is also count data, just like the number of clutch caring events.

All figures with categorical predictors (e.g., "treatment"): It would be useful to include some type of asterisk system (e.g., * = P < 0.05, ** = P < 0.01, NS = P > 0.05…) for delineating P values in these figures. Especially in Figure 5, where 5C is significant (P = 0.028) and 5D is not (P = 0.089), the reader cannot see these differences at a glance. The authors do something like this in Figure 2, but not in other figures. Of course, we can discuss issues about whether P values really represent biological significance, but since that's the framework we're working under here, it would be useful to reflect in the figures.

Lines 332-336: Though I think this point is likely correct, the authors did not actually show how the experiment impacted the later dominance and reproductive success of the offspring. Therefore, I might suggest revising this to "…outgroup conflict likely negatively impacts the fitness of multiple generations…."

Methods lines 534-536: This is incredibly nit-picky, but I'm mostly curious. Why were there such varying numbers of eggs removed for the lipid versus protein analyses?

*Reviewer #3 (Recommendations for the authors):*

I enjoyed reading this paper again, and the authors now do a much better job of explaining the experimental set-up prior to the results. The only thing I still think is missing is a brief description (prior to the results) of how the animals behave during the experimentally induced conflicts and a definition of what "defensive actions" involve. A few small additions should suffice, and add important context. I would suggest promoting the descriptions of the behaviours coded as defensive (headbutting, displays, etc) from L514 to the results paragraph starting at L129. It would also be good to hear a little more about the behaviour of the "intruder". Did they frequently antagonise their neighbours, or exhibit aggressive displays or attacks?

Otherwise, the authors have satisfactorily addressed my comments and, for my part, I see no reason that this paper shouldn't be published – it is very interesting!

---

## [Author Response]

Essential revisions:1) Eliminate stepwise model selection procedure to minimize the risk of false positives. Report either betas and p-values from the full model, or via removal of the effect of interest from the full model, followed by comparison of model fit (e.g., using likelihood ratio tests).

We have revised our statistical analyses as requested. We have removed only non-significant interaction terms from our models (as per Enqvist 2005 and the suggestion of Reviewer #2 below), before reporting the statistical output from the remaining final models (lines 665–668). All factor significance assessments were done using likelihood ratio tests (LRTs) (lines 663–665). Throughout the main manuscript, we report parameter estimates and associated 95% confidence intervals, and the results of LRTs for treatment and, where significant or near-significant, its interaction with treatment duration (lines 144–147). When an interaction between treatment and treatment duration was found to be statistically significant, we further present the parameter estimates, confidence intervals and LRTs for each treatment separately (lines 668–670). Full outputs from analyses are presented in the Supplementary Tables (lines 147–148).

2) Explain the experimental paradigm early in the manuscript to facilitate interpretation by readers. Key points include the fact that the intrusion paradigm involved no actual physical contact, and that in Experiment 2, all clutches prior to Day 13 were destroyed. Clearly motivate the analyses that are later presented in the Results section by reference to description of the experimental paradigm.

We have included explanation of the experimental paradigm at the start of the Results section (line 114ff.), including key requested information that there could be no physical contact between focal groups and intruders (lines 130–132), and that in Experiment II clutches prior to Day 13 were removed (lines 141–144). We have included the reason for each of the experiments, in terms of what response measurements were obtained in each case (Experiment I: lines 135–137; Experiment II: lines 139–141).

3) Intergroup conflict effects on the number of surviving offspring, the closest proxy to fitness used here, are one of the most important results of the paper. If possible, present a straightforward comparison of this outcome for the treatment versus control groups across the duration of the experiment, along with a simple analysis of the magnitude of this difference (e.g., using a paired t-test). Clarify the degree to which conflict effects on surviving offspring are magnified over time (e.g., in terms of the % of additional surviving offspring in the control group, across a given number of days since the start of the experiment).

In redoing our analyses (see response to point 1 above), the storyline relating to number of offspring surviving became simpler: the absolute number of offspring surviving decreased with treatment duration in the Intruded treatment but not in the Control treatment (lines 260–265). This provides a very straightforward comparison (see Figure 5A). [NB It is not possible to use a simple t-test because several groups produced multiple clutches with offspring surviving to 1-month, hence our use of mixed models.] We have provided a simple indicator of the reduced offspring survival likelihood in Intruded groups cf. Control groups across time (lines 265–267).

4) Pay close attention to the reviewer recommendations on statistical analysis below, especially consistency in declaring effects "significant"/"near significant" and reporting of "near-significant" effects.

We have made sure that we are consistent in the presentation of results. We have made it clear that in the main text we are presenting the main treatment effect (whether significant or not) and, when significant or near-significant, the interaction between treatment and treatment duration (lines 144–147); these are the factors of core interest. All other factors in models are those that we are controlling for and so are provided in full in the Supplementary Tables (lines 147–148).

Reviewer #2 (Recommendations for the authors):– To address my concern about stepwise model reduction, the authors should report the significance of each effect by comparing the fit of a full model to that of a model without that effect. A second option would be to remove only non-significant interaction terms, if the goal is mainly to investigate the fixed effects (Enqvist 2005 An. Beh., but see Shielzeth 2010 Meth. Ecol. Evo.). I often take the second approach in my own work, but am realizing that the first approach may be more appropriate.

We have taken the second approach suggested by the reviewer, because our main terms of interest were the factor ‘treatment’ and its interaction with the ‘treatment duration’, and thus we wanted to be able to assess to effect of treatment on its own, when interactions involving treatment were found to be not significant. We have explained this approach in the Methods (lines 663–670).

– To address my concern about a lack of clear hypotheses, the authors could outline why each specific metric was measured and why each relates to fitness early on (e.g., at the beginning of each paragraph or at the end of the Introduction).

We have included a new paragraph at the end of the Introduction (line 86ff.) providing explicit reasons for specific predictions relating to our response variables.

Comments below are line-by-lineTitle: "outgroup conflict" might be a less widely-used term than "intergroup conflict" in this literature. For example, the titles of most of the first 10 references in this paper use "intergroup" instead of "outgroup". Consider using "intergroup" instead of "outgroup".

In our view, ‘outgroup’ conflict encompasses all conflicts involving the whole or some members of a focal group, and outsiders, whether they be a group or a single individual. We consider ‘intergroup’ conflict as a subset of outgroup conflict, in which two groups are in conflict. We have added an explanation of our terminology (lines 40–42). As our intruder was a single individual, we believe the term *outgroup* is the most appropriate and so have left that in the title.

– Line 43: consider citing Green et al., 2020, which is a review of intergroup contest studies and how they focus on factors like the determinants of intergroup contest success. Green PA, Briffa M, Cant MA. 2020. Assessment during intergroup contests Trends Ecol Evol. 36(2):139-150.

We have cited this review (line 47).

– Line 47: consider citing Preston et al., 2020, which looked at short-term and longer-term (3 days) changes in grooming and aggressive behavior within groups after simulated intergroup conflict. Preston EFR, Thompson FJ, Ellis S, Kyambulima S, Croft DP, Cant MA. 2020. Network‐level consequences of outgroup threats in banded mongooses: Grooming and aggression between the sexes. J Anim Ecol.

We have cited this paper (line 53).

– Lines 54-55: I had trouble telling whether there was a timescale difference in these three studies. Did the studies truly differ in a focus on "foetal survival" (first two) versus "infant survival" (third)?

We believe that there is a difference between these studies. Our understanding is that all three of them considered (at least as part of their studies) the impact of intergroup conflict during pregnancy (which is the focus of our sentence; lines 60). Whereas Kerhoas et al., 2014 and Thompson et al., 2017 found positive correlations of prenatal intergroup conflict with foetal survival, Lemoine et al., 2020 did not obviously consider that response measure but did find a negative correlation of prenatal intergroup conflict on inter-birth intervals and post-birth (infant) survival. So, we have left this sentence as is.

– Lines 56-57: It would be useful to have some more detail about why we really need these experiments. What is missing from those observational studies that an experimental study could really help us learn? Put another way, can you give more details on the importance of this "causal link"?

It is a fundamental tenet of scientific research that correlations found in observational data could arise due to confounding effects. For instance, a positive correlation between outgroup conflict and improved foetal survival might be because groups with better-quality territories engage in more interactions with neighbours (who are after those resources) but those resources mean that mothers are in better condition; outgroup conflict per se is not necessarily driving the reproductive effect seen. Experiments are needed to test causality because they can isolate the effect of a particular variable (in this case, outgroup conflict). We have added a sentence to the Introduction explaining the general need for experiments (lines 64–66).

– Lines 88-96: It's unclear to me from these results and Figure 2 why a Wilcoxon test versus a t-test was used for certain tests. My assumption is that this relates to the normality of the data distributions. Perhaps a simpler way to get at the same question would be to build a linear mixed model predicting # of defensive behaviors from treatment, time, and individual type (dominant male/female, subordinate)?

Yes, we chose the most appropriate statistical test (parametric vs non-parametric) depending on consideration of the relevant assumptions (lines 638–639). We were specifically interested in whether any or all categories of individuals had shown signs of having habituated to our simulated intrusions. The fact that none of the categories of individuals decreased their defensive contributions—indeed, both dominants increased their defensive actions towards the intruder—shows that groups did not habituate to our treatment and that they continued to perceive the intruded in their territory as a threat (lines 158–159). Using a linear mixed model with a three-way interaction term would not, in our view, be a simpler way of considering this; since this is also not the core aspect of the paper, but rather a check that intruders were indeed viewed as a threat throughout the experiment, we would rather leave this as is.

– Results in general: I'm a bit confused as to why significant effects are reported for certain interaction terms, and then also for each level of the interaction. For example, lines 180-184 detail the significant interaction (and associated P-value from a Chi-squared test), as well as reporting t-values and P-values from each level of the interaction (e.g., treatment duration effects within each treatment). I believe just reporting the P-value from the Chi-squared test for the interaction term shows that the interaction is significant, which should then be followed by simpler reporting of the estimate for each level of the interaction. It seems unnecessary to test whether each level of the interaction is also significant.

Testing each level of treatment is key to interpreting the results of the main analyses; without testing the treatments separately following a significant treatment*treatment duration interaction, it is not possible to state conclusively whether and how the simulated intrusions impacted the groups over time. For instance, while egg volume increases over time in the Control treatment but remains stable in the Intruded treatment (Figure 3C), egg protein content remains stable in the Control treatment but decreases significantly over time in the Intruded treatment (Figure 3D). Reporting just a significant interaction does not by itself describe the nature of the effect.

– Results in general: There are several examples where the authors report "trending" or nearly-significant results that match their predictions (e.g., lines 161-163; 193-194; 216-217; 266). However, there are also instances in the reporting of the statistical models in which similarly trending factors were removed from statistical models (e.g., Table S2; S4; S6; S8; S9). I believe some of these removed values were reported in the results (e.g., line 266 matches to Table S9). However, it is confusing to understand why certain terms were removed from the statistical models, especially if they are reported in the Results (i.e. we can assume the authors consider these are important). See above for comments regarding statistical analyses.

In redoing our analyses in light of earlier comments, this issue has been resolved (see earlier responses).

– Figure 3 and all scatterplots: are these lines showing best fit from the LMM, i.e. the estimates from the LMM? Or are they plotting simply using a function like geom_smooth in ggplot? Showing the actual predicted estimates from the model on the scatterplot would give the reader a better idea of both the raw data (from the points) and the model estimates (from the lines and associated ribbons of confidence intervals). There are also methods to do this for boxplots. All of these methods can be accomplished using the "effects" package in R.

We have replotted all result figures in light of the feedback. We plotted figure 2 panels using the package “ggPaired” that allowed us to connect the datapoints from the same group/individual. Where the treatment-by-treatment duration interaction was significant or near-significant, we used the package “interactions” to plot predicted means with associated 95% confidence intervals alongside the raw data (Figure 3B–D, Figure 4A, B, Figure 5B) or partial residuals (Figure 5A). To visualise main treatment effects (or lack thereof), we used the package “jtools” to plot predicted means and associated 95% confidence intervals alongside the raw data (Figure 3A, Figure 4C, Figure 5C, D).

– Lines 166-168: This is a bit confusing, because I thought the protein and lipid analyses were only conducted in Experiment II on eggs from the first clutch? So, how can conclusions about protein/lipids in the eggs from later clutches be made if only the first clutch was analyzed? Or, is this because clutches were analyzed across groups, i.e., several clutches for a given focal group were not analyzed, but across the entire experiment the first clutch from each group gave a time effect?

We have resolved this confusion by clearly delineating results arising from Experiment I and Experiment II, and not combining them in the same sentence. It is correct that protein and lipid analyses were only from Experiment II (lines 139–141); these results can be found in a Results paragraph relating only to that Experiment (line 203ff.).

– Figure 4B: There appears to be an outlier with an incredibly high number of cleaning events at approximately 60 days treatment duration in the Intruded group. It would be useful to know if there are any biological differences between this observation and the others. Also, if this is a statistical outlier, it would be useful to know whether removing it affects the results. While I don't suggest the removal of statistical outliers simply because they're outliers (i.e., there should be a biological reason for data to be removed), it's important to take not of any outlier values like this.

There was nothing noteworthy about this group or this particular clutch to justify its removal from the dataset: it was the third clutch produced by the female during the study, was the 8^th^ largest clutch recorded and had the 13^th^ highest hatching success; only three fry survived to 1 month (so no behavioural data were collected), but they were of average size.

– Lines 219-224: I might be being dense here, but I'm unsure of what this sentence is saying. How is the relationship (or lack thereof) between the number of hatched fry and the number of offspring reaching 1 month of age not offspring survival? It seems from Table S7 that there was a difference between offspring survival to 1 month and survival from hatching to one month. I guess I'm having trouble understanding why we need both of these measures.

We agree that including both these measures was superfluous and somewhat confusing. In redoing our analyses in the ways suggested (see above), a simpler story emerged in relation to offspring survival. We thus now simply present one, clear-cut analysis (lines 260–267) and associated figure (Figure 5A).

– Lines 257-259: Could another interpretation of the return to activity data be regarding boldness? That is, offspring from Intruded groups were bolder later on in the experiment?

In principle, repeated territorial intrusions could lead to the production of increasingly bolder offspring. However, we did not find any complementary evidence for boldness (e.g., greater pre-stimulus activity levels or increased nearest-neighbour distances) so do not think there is strong justification for including this possibility.

– Lines 277-280: However, your results are counter to those of other studies you described in the Introduction, which found no or positive effects of intergroup conflict on reproduction and survival. Consider re-reporting that work here, as well, for consistency and clarity.

We have included explicit mention of those other studies at this point in the Discussion (lines 320–322).

– Paragraph lines 311-340: I'm a bit confused as to the point of this paragraph. I understand the authors may want to compare the pros and cons of their experimental approach to a more field-based, correlational approach. However, the result of this comparison mainly convinces me that the experiment under-stressed the study animals, as compared to what they might experience in the wild. Further, the reporting of the lack of impacts on adult body mass, egg mass, etc… essentially make the point that the treatment did not affect groups much. This is counter to the rest of the paper's point, showing that groups were significantly affected. Do the authors have any hypotheses as to why they found specific results for the variables they did (e.g., egg protein, but not egg mass)? Do the authors have any thoughts or data on the relative reproductive success of these experimental groups compared to groups in the wild? I.e., are these experiments actually replicating stressors in the wild?

We have simplified the message here. The core point of our paper is that the experimental tests showed various impacts of outgroup conflict on reproduction, ultimately resulting in fewer and smaller young surviving to 1 month of age. Whilst our experimental intrusions were possibly longer than would be seen in the wild, there are several reasons to expect greater effects in the wild (lines 373–375). Ultimately, future studies would beneficially conduct wild-based experiments, although those are logistically very challenging and have important ethical considerations (lines 390–392).

Methods line-by-line– Lines 408-411: If one of the points of the present experiment was understanding the effects of short- versus long-term intergroup conflict, why destroy clutches produced before three intrusions? Wouldn't these give valuable information on short-term effects (or the lack thereof)?

At the outset, our main aim was comparing reproductive rate, investment and output between the two treatments (Intruded and Control). So, we decided to remove the clutches produced before at least three intrusions had occurred because we assumed groups would not have been substantially affected by our treatment in that period and would have reduced our ability to assess the effect of treatment on response variables such as latency to spawn. In analysing our data, we sometimes found that the interaction between treatment and treatment duration was important, but this had not been the core focus when we set up the experiments. Only three clutches were destroyed in Experiment I (line 458–461)—two on the morning of the first day, before the groups had experienced any intrusions, and one on the second day, before the group has experienced its second intrusion—so there is likely no great loss in terms of understanding.

– Line 549: was the number of offspring in the experiment considered as a factor predicting responses? In Table S8, I believe "number of offspring" is used as a fixed effect, but I was unclear as to whether that was number of offspring produced, or number used in the startle experiment. One might expect that the experimentally-designed groups with 10 offspring were different from those with 5.

Yes, the factor “number of offspring” refers to the number of individuals used in the test, and not the number of surviving offspring. We have clarified this point in Supplementary Table S8 by using the term ‘number of offspring in test’.

– Lines 550-551: Was there any sand in the bottom of this startle tank? I would imagine that offspring sinking responses, if meant to blend into the substrate, would be different if there were no substrate to blend in to.

No, we did not add sand to the tank to facilitate the cleaning of the tank and removal of olfactory cues from other broods that could have influenced the focal individuals’ behaviour during the startle test. It is possible that the lack of sand could have influenced the offspring behaviour, but the vast majority of individuals did display sinking behaviour in response to the startle stimulus, and we expect that the lack of sand in the bottom of the tank would have affected offspring from Control and Intruded groups alike.

– Lines 585-589: Stepwise model reduction has come under heavy criticism in the past 15 years. See, for example, Forstmeier and Schielzeth (2011), Mundry and Nunn (2009), and Harrison et al., (2018). This process can greatly increase the likelihood of false-positive results, and can inflate parameter estimates. From my knowledge, removal of non-significant interaction terms is OK, if one is interested in achieving accurate parameter estimates for main effects. However, it is rarely advised to remove main effects. Ideally, the authors would report significance tests from the full model, or use information-theoretic approaches. However, the authors could consider removing only non-significant interaction terms, as a means of getting better estimates for the main effects (Engqvist 2005).Forstmeier W, Schielzeth H. 2011. Cryptic multiple hypotheses testing in linear models: Overestimated effect sizes and the winner's curse. Behav Ecol Sociobiol. 65(1):47-55.Mundry R, Nunn CL. 2009. Stepwise model fitting and statistical inference: Turning noise into signal pollution. Am Nat. 173(1):119-123.Harrison XA, Donaldson L, Correa-Cano ME, Evans J, Fisher DN, Goodwin CED, Robinson BS, Hodgson DJ, Inger R. 2018. A brief introduction to mixed effects modelling and multi-model inference in ecology. PeerJ. 6:e4794.Engqvist L. 2005. The mistreatment of covariate interaction terms in linear model analyses of behavioural and evolutionary ecology studies. Anim Behav. 70(4):967-971.

Thank you for the advice and suggested literature. We agree with the reviewer and have revised our analyses accordingly: we removed only non-significant interactions (lines 665–668), allowing us to assess the effect of treatment more accurately when the factor was not involved in significant interactions (Supplementary Tables S1–S9).

– Lines 603-607: I would appreciate learning more about this approach from the authors. It's very unclear to me why you wouldn't simply fit an appropriate response variable distribution in the initial model fitting step. For example, why not just use a Poisson distribution for the number of caring events response variable? This reads as though all response variables were forced into a Gaussian distribution. Other distributions are appropriate for other types of data, and do not require this step. I might be misunderstanding, however.

We agree with the reviewer that other distributions may be more appropriate for some datasets and have revised out analyses accordingly. To this effect, the number of caring events has been analysed using a GLMM with a negative binomial distribution, and time spent caring was analysed using a GLMM with a Gaussian distribution and log-link function (lines 650–655); results were qualitatively the same as before. For latency to spawn, the GLMM with a γ distribution had computational issues that could not be fixed so we log_10_-transformed the raw data before using a LMM (lines 649–654). Formal comparison of possible models using the package “performance” confirmed that our chosen method performed better than alternatives.

– Lines 609-612: I'm unclear as to why these clutches with zero hatching success were removed. The reasons these clutches had zero hatching success seem incredibly relevant to the study: if a clutch was eaten it seems evidence of cannibalism by the focal group (see lines 227-230 for a statement relevant to cannibalism in this species). If the clutch had a fungal infection, it would seem that the focal group's lack of care could have resulted in an infection.

We agree and have included all clutches in our analyses; the revised results are qualitatively similar to the previous results (line 524, Table S6).

Reviewer #3 (Recommendations for the authors):My recommendations are superficial. My main comment is that I was frustrated by the cursory description of the experimental set up prior to the results. I know the eLife format is for the methods to go at the end of the manuscript but I really do think that at the very least the authors ought to explain how the simulated intrusions were enacted. The results didn't really come in to focus for me until after I had read the methods.Line comments:L70: on natural behaviour in the lab: have multiple groups been kept in the same tank and formed territories? If so, would be worth mentioning.

No, groups were kept in separate tanks in both experiments (lines 114–118, 410).

L84: This is where I would like to know how these territorial intrusions were simulated.

We have included information on territorial intrusions (lines 130–135).

L100: In general, the paper is clearly written. However, I found this paragraph confusing and had to re-read several times. The description of the experimental set-up in the methods, in contrast, reads very clearly.

We have included more, and clearer, information about our methods early in the Results (line 114ff.).

L126: '40%'. Can you give a 95% CI here?

This value is the relative difference between the treatment means and so there is not a 95% CI value to provide as well.

L423-441: This is the critical detail on how the incursions were set up that I would have liked to see in the main text.

We have included this information in the Results (lines 130–135).

[Editors’ note: further revisions were suggested prior to acceptance, as described below.]

The manuscript is much improved and close to publication-ready. However, the reviewers offered a few suggestions that may be helpful in communicating with your audience, so we are returning the draft to you to address these comments at your discretion.1. Addressing Reviewer 2's notes about consistent reporting of significant parameter estimates.

We explicitly state in lines 149–152 that we report in the main text only the effect of treatment and, where significant or near-significant, the effect of the treatment by treatment duration interaction, because these are the focus of our work. See below for full response to Reviewer 2.

2. Clarifying the set of behaviors coded as "defensive action".

We have added a description of the classes of behaviours coded as “defensive action” in lines 133–137, as well as a brief explanation of the types of behaviours typically adopted by intruders.

Reviewer #2 (Recommendations for the authors):Overall, I think the authors have done a great job of addressing my and the other reviewers' concerns. I am glad that the statistical suggestions have helped simplify the message, which is still an engaging and exciting one. I also greatly appreciate the laying out of predictions and the details of the methods early on. This will hopefully help readers get oriented. I have one main point, which I lay out first, and then a few line-by-line comments. I don't believe any of these comments are likely to change the main message of the manuscript.Main comment:On line ~680-681 in the Methods, the authors say they "provide significant parameter estimates…in the main text." However, looking at the Supplemental Tables in comparison to what is reported in the main text, only some significant effects are reported in the main text, not all. I'm curious as to why this is. For example, in the analysis of inter-clutch interval (Supp Table 1), female size had a highly significant effect (P = 0.017) but only the effect of treatment was discussed in the Results (lines 180-181). Similarly, in the analysis of egg investment, only the near-significant effect of the interaction between treatment and treatment duration was discussed (lines 195-197), but not the significant fixed effect of treatment. Also, an effect of clutch size on protein content and an effect of treatment x female size on protein content (Supp Table 3) were unreported in the Results. If these significant, but unreported, effects are not of interest, then one might ask why they were included in the model in the first place (aside from where necessary, like the treatment main effect when there is an interaction of interest). Or, are these omissions due to space constraints? There is likely a simple way to include a fuller reporting of these results. E.g., around line 195 the authors could report that "…there was a significant effect of treatment duration on clutch size (STATS HERE). This was likely driven by the interaction between treatment and treatment duration, in which Intruded females…." Or, an alternative would be in the Methods to state clearly that the authors only report selected results relevant to their hypotheses, but that some significant findings are reported only in the Supplementary Tables.

We explicitly state in lines 149–152 that we only provide in the main text the effect of treatment and, where significant or near-significant, the effect of the interaction between treatment and treatment duration, because these are the factors of core interest. We have included in the additional sentence there that the Supplementary files contain full model outputs including all tested variables, whether significant or not (lines 152–153). We have amended the sentences in the Methods to clarify that only those core results are detailed in the main text and that all model results are detailed in the Supplementary files (lines 676–680). Other factors such as female size and clutch size are included in some models because these traits are known to affect some response variables. Their inclusion allows us to assess the effect of our treatment above and beyond their expected influence on the response variable, but any discussion of such significant confounding effects would be tangential to our main story and thus we have not included it in the main text.

Line by line:Lines 103 and 107: For the other predictions the authors make directional predictions, e.g. a negative effect. Can they specify the direction of the effect here? Or, are there no directional predictions?

While we generally predicted a decrease in reproductive investment and success due to outgroup conflict, we did not have directional predictions for each of these variables due to the potential for behavioural and physiological trade-offs to occur.

Lines 132-151: These details are incredibly helpful! Thanks for including them.

Thank you for the original suggestion to do so.

Figure 3D: There are two red points at the end of treatment duration, right in the path of the line for the control (blue) effect. Are these points miscolored? It could just be an incredible chance that the line for control goes right through these intruded points, but I would also expect the 95% CI ribbon for intruded to be wider if these points are from the intruded treatment.

The datapoints are for the Intruded treatment and the 95% CI ribbon is correctly estimated.

Supp Table 4A, Figure 4A, and lines 656-663: I could be wrong, but I would imagine that the number of clutch visits analysis should also be done using negative binomial error distribution, or a Poisson distribution. It is also count data, just like the number of clutch caring events.

Thank you for pointing this out. We have replaced it with a negative binomial GLMM, which produced very similar results. We have revised Table S4 and figure 4A, the Results text (lines 242–245) and the Statistical Methods (line 656), accordingly.

All figures with categorical predictors (e.g., "treatment"): It would be useful to include some type of asterisk system (e.g., * = P < 0.05, ** = P < 0.01, NS = P > 0.05…) for delineating P values in these figures. Especially in Figure 5, where 5C is significant (P = 0.028) and 5D is not (P = 0.089), the reader cannot see these differences at a glance. The authors do something like this in Figure 2, but not in other figures. Of course, we can discuss issues about whether P values really represent biological significance, but since that's the framework we're working under here, it would be useful to reflect in the figures.

We have added p-values to all panels in Figures 3–5, in line with Figure 2.

Lines 332-336: Though I think this point is likely correct, the authors did not actually show how the experiment impacted the later dominance and reproductive success of the offspring. Therefore, I might suggest revising this to "…outgroup conflict likely negatively impacts the fitness of multiple generations…."

Done (line 332).

Methods lines 534-536: This is incredibly nit-picky, but I'm mostly curious. Why were there such varying numbers of eggs removed for the lipid versus protein analyses?

The number of eggs collected for the different analyses was based on literature searches and consultations with colleagues who have used the same protocols, albeit on different taxa.

Reviewer #3 (Recommendations for the authors):I enjoyed reading this paper again, and the authors now do a much better job of explaining the experimental set-up prior to the results. The only thing I still think is missing is a brief description (prior to the results) of how the animals behave during the experimentally induced conflicts and a definition of what "defensive actions" involve. A few small additions should suffice, and add important context. I would suggest promoting the descriptions of the behaviours coded as defensive (headbutting, displays, etc) from L514 to the results paragraph starting at L129. It would also be good to hear a little more about the behaviour of the "intruder". Did they frequently antagonise their neighbours, or exhibit aggressive displays or attacks?

We have added this information as requested (lines 133–137). Intruders often respond with aggressive postures and occasional attacks, but also with submissive postures and displays, and with evasive behaviour. We did not quantify these behaviours for the intruder and therefore cannot comment on the frequency of the different behaviours.